# Temporal superposition and feature geometry of RNNs under memory demands

**Pratyaksh Sharma**[*]
Imperial College London

**Alexandra M. Proca**[*]
Imperial College London

**Lucas Prieto**
Imperial College London

**Pedro A.M. Mediano**
Imperial College London
University College London

## Abstract

Understanding how populations of neurons represent information is a central challenge across machine learning and neuroscience. Recent work in both fields has begun to characterize the representational geometry and functionality underlying complex distributed activity. For example, artificial neural networks trained on data with more features than neurons compress data by representing features non-orthogonally in so-called *superposition*. However, the effect of time (or memory), an additional capacity-constraining pressure, on underlying representational geometry in recurrent models is not well understood. Here, we study how memory demands affect representational geometry in recurrent neural networks (RNNs), introducing the concept of temporal superposition. We develop a theoretical framework in RNNs with linear recurrence trained on a delayed serial recall task to better understand how properties of the data, task demands, and network dimensionality lead to different representational strategies, and show that these insights generalize to nonlinear RNNs. Through this, we identify an effectively linear, dense regime and a sparse regime where RNNs utilize an interference-free space, characterized by a phase transition in the angular distribution of features and decrease in spectral radius. Finally, we analyze the interaction of spatial and temporal superposition to observe how RNNs mediate different representational tradeoffs. Overall, our work offers a mechanistic, geometric explanation of representational strategies RNNs learn, how they depend on capacity and task demands, and why. [1]

## 1 Introduction

A major goal in both machine learning and neuroscience is to understand how populations of neurons represent information, and why certain representational geometries are preferred in different settings. Characterizing different strategies can lead to increased interpretability of artificial neural networks (ANNs), as well as help us better understand the functionality of certain brain regions.

A prevalent theme in neuroscience has been the pursuit of highly-specialized functional components in the brain. Much classical work aimed to demonstrate the functions of individual neurons (Yuste, 2015), such as simple and complex cells in the primary visual cortex (Hubel & Wiesel, 1962) or place cells in the hippocampus (O'Keefe & Dostrovsky, 1971). More recently however, large scale population recordings have revealed that neural computation is often distributed across many neurons (Yuste, 2015; Saxena & Cunningham, 2019; Ebitz & Hayden, 2021). It's thought that such coding strategies are important for flexible, complex behavior (Fusi et al., 2016; Tye et al., 2024).

A parallel line of work has investigated representational geometry in ANNs. While earlier work tried to develop models where neurons have specialized functions (disentangled representations or monosemantic neurons) (Bengio et al., 2014; Olah et al., 2020; Cammarata et al., 2020), recent work has focused on *superposition*, a phenomenon that characterizes a class of distributed representations.

In particular, ANNs are often challenging to interpret due to *polysemanticity*, where neurons respond to many unrelated mixtures of inputs. One explanation for polysemanticity is the superposition

---

[*]Co-first author; correspondence to `sharma.pratyaksh@gmail.com`, `a.proca22@imperial.ac.uk`.

[1]Code available at `https://github.com/kashparty/iclr-rnn-superposition`

hypothesis (Elhage et al., 2022), which posits that ANNs utilize a compression strategy where they represent more features (e.g., directions in activation space) than there are neurons by allowing some interference (non-zero dot products) between them. Superposition is particularly effective when features are sparse, although it leads to features no longer mapping onto individual neurons. In the era of big data, even in models with millions of parameters, superposition appears ubiquitous (Bricken et al., 2023; Templeton et al., 2024) – an effective strategy arising from the inherent sparsity of features in real-world data. A major focus of mechanistic interpretability is now finding new techniques to extract such features from superposition. Fewer studies, however, have focused on understanding the feature geometry induced by superposition (Sharkey et al., 2025).

Previous work in interpretability has not considered the effect of *time* (or memory) as a capacity-constraining pressure. While less of an issue in feedforward and transformer architectures, this pressure does arise with recurrence. Recurrent neural networks (RNNs) are important for learning tasks with temporal dependencies and models of dynamical systems. They are commonly used as cognitive models in neuroscience (Barak, 2017), as they often replicate the neural activity of animals when trained on similar tasks (Vyas et al., 2020). Additionally, RNNs are increasingly popular for long-range sequence modeling due to their computational and memory efficiency (Gu et al., 2022b; Orvieto et al., 2023; Gu & Dao, 2024). Understanding the effect of capacity through time on representational geometry is important, as it affects maintenance of long-range dependencies and could have implications for cognition with limited resources, such as working memory.

In this work, we study how time, or task (memory) demands, affects representational geometry and capacity in RNNs through the lens of feature superposition. We develop a theoretical framework to better understand representational strategies employed by RNNs and how data and network dimensionality interacts with memory demands. Our results characterize and explain behavior across different recurrent architectures and task settings, providing insight into what RNNs learn and why. Our contributions are as follows:

- We introduce the concept of temporal superposition in RNNs.
- We distinguish two forms of interference – projection and composition interference – and show how they impact behavior.
- We derive an expression for the loss on a simple recall task that decomposes into four interpretable terms, allowing us to explain the geometric strategy employed by the RNN.
- We study the learning dynamics of RNNs in terms of changing representational geometry.
- In RNNs with ReLU nonlinearities, we identify the existence of an interference-free space into which many feature directions can be tightly packed.
- We identify a phase transition in the geometry between the dense and sparse regimes.
- We study the interaction of spatial and temporal superposition, and how RNNs mediate this tradeoff depending on task requirements.

## 2 KEY INTUITIONS: SPATIAL & TEMPORAL SUPERPOSITION

Here we introduce the key ideas we will develop formally in the remainder of the paper. First, we review *spatial* superposition, which was studied extensively by Elhage et al. (2022) in feedforward networks, and then extend these concepts to characterize *temporal* superposition in RNNs.

**Spatial superposition.** We follow the linear representation hypothesis and assume that features (loosely defined as interpretable properties of the input; see Elhage et al. (2022); Park et al. (2024)) are represented as directions in activation space. When there are more features of the data than neurons and features are sparse (i.e., do not commonly co-occur), it becomes optimal for neurons to represent features non-orthogonally (in shared dimensions of the activation space) so as to compress more in (Figure 1a). This comes at the cost of possible interference between co-occurring features, but if features are sufficiently sparse, this can be outweighed by the benefit of representing more features. Spatial superposition only becomes viable in the presence of a nonlinearity that reduces interference between features; linear networks instead learn a PCA of the most important features.

**Temporal superposition.** In addition to spatial superposition, we claim that RNNs exhibit another form of superposition due to the axis of time and that this phenomenon is fundamentally different

(a) Spatial superposition     (b) Temporal superposition

(c) Projection interference

(d) Composition interference

Figure 1: **Spatial and temporal superposition both occur in RNNs and are characterized by different representational strategies. (a)** When there are more input features (5: A-E) than hidden dimensions (2) and features are sparse, it becomes favorable to compress these features into the activation space non-orthogonally in spatial superposition. **(b)** When an input feature (A) must be held in memory for more time (red: 0 timesteps old; green: 5 timesteps old) than hidden dimensions (2) and features are temporally sparse, it becomes favorable to utilize temporal superposition to exploit the interference-free space opposite the output feature. **(c)** Projection interference (*red dashed line*) occurs when the activation of a feature (*black*) is read-out (*blue arrow*) as the activation of a different feature (*gray*). **(d)** Composition interference (*red dashed line*) occurs when the activation of multiple features (*black*) is linearly combined into an activation that imitates another feature (*gray*).

from the feedforward case. In particular, in addition to having some spatial component, we can think of each input feature as also having a temporal component dependent on its sequential position. This means that features are represented differently depending on which timestep they occur at, even if the input itself is the same. In other words, representations of features are now determined by "when" just as much as "what". For example, if an RNN receives an impulse of some feature A as input at timestep $t$, the representation of this input will move through a set of distinct feature directions as time goes on and the impulse grows older, until it ceases to be task-relevant (as in Figure 1b). If features are held over longer periods of time (because they remain task-relevant), more feature directions are compressed into the hidden state. The hidden state therefore acts as a bottleneck, such that RNNs are forced to either forget features or represent more features than dimensions (in superposition) as the length of the task-relevant input window increases. In addition to compressing more features due to memory demands, features must also be read out at particular timesteps, rather than being immediately available as in feedforward models, giving rise to different behavior from that of spatial superposition. We distinguish *temporal* superposition as resulting from representing features across a longer period of time (higher memory demand) in a lower-dimensional activation space, whereas we refer to *spatial* superposition as representing more input features (higher-dimensional data feature space) in a lower-dimensional activation space.

## 3 MATHEMATICAL SETUP

### 3.1 RNN MODEL

We study a RNN parameterized by matrices $W_x \in \mathbb{R}^{N_h \times N_x}, W_h \in \mathbb{R}^{N_h \times N_h}, W_y \in \mathbb{R}^{N_h \times N_y}$ with a hidden state $\boldsymbol{h}_t \in \mathbb{R}^{N_h}$ that receives an input $\boldsymbol{x}_t \in \mathbb{R}^{N_x}$ at each timestep $t$ and produces an output $\hat{\boldsymbol{y}}_t \in \mathbb{R}^{N_y}$. The RNN is given by

$$\boldsymbol{h}_t = W_x \boldsymbol{x}_t + W_h \sigma_h(\boldsymbol{h}_{t-1}) \quad \hat{\boldsymbol{y}}_t = \sigma_y(W_y^\top \boldsymbol{h}_t) \tag{1}$$

where $\sigma$ represents either a linear activation or ReLU, depending on the setting we consider. We refer to the model with linear recurrence and readout as a linear RNN, the model with linear recurrence and nonlinear readout as a state space model (SSM), and the model with nonlinear recurrence and readout as a nonlinear RNN. We first focus on the setting where both $\sigma_h, \sigma_y$ are linear and later consider cases of nonlinearity. We initialize the hidden state $\boldsymbol{h}_0$ at $\boldsymbol{0}$. In the case where $\sigma$ is linear, this yields

$$\boldsymbol{h}_t = \sum_{i=1}^{t} W_h^{t-i} W_x \boldsymbol{x}_i \tag{2}$$

## 3.2 FEATURE DIRECTIONS

We define $s := t - i$ to indicate how many 'recurrences' an input from timestep $i$ has undergone at timestep $t$ (or the 'time window' it has been in the RNN for). This allows us to account for a feature in terms of how long it has been in the RNN relative to a particular timestep. Letting $W_s := W_h^s W_x$, we can rewrite the expression for the hidden state above as

$$\boldsymbol{h}_t = \sum_{s=0}^{t-1} W_h^s W_x \boldsymbol{x}_{t-s} = \sum_{s=0}^{t-1} W_s \boldsymbol{x}_{t-s} \tag{3}$$

To isolate the effect of temporal superposition and make visualization easier, we first consider scalar inputs and outputs ($N_x = N_y = 1$) and a two-dimensional hidden size ($N_h = 2$). After building a thorough understanding of temporal superposition, in Section 4.5 we introduce spatial superposition by increasing the dimensionality of the data ($N_x = N_y = 5$) and study its interaction with temporal superposition. We note that our framework and main results naturally extend to multidimensional inputs and outputs, and higher dimensional hidden spaces (Appendix G). For clarity, we refer to $W_y, W_s \in \mathbb{R}^{2 \times 1}$ as $\boldsymbol{w}_y, \boldsymbol{w}_s$ to indicate that they are vectors, and to $\boldsymbol{x}, \boldsymbol{y}, \hat{\boldsymbol{y}} \in \mathbb{R}^1$ as $x, y, \hat{y}$ to indicate they are scalars (the equations above remain otherwise the same).

In this form, we can see how each input feature $x_{t-s}$ is independently and linearly represented in the hidden state in the direction given by $\boldsymbol{w}_s$ (the feature direction). This makes it clear that, at time $t$, the model has access to the entire history of $t$ input features, but that for $t > N_h$, the hidden state inevitably becomes bottlenecked. Further, this form illustrates the role of the readout $\boldsymbol{w}_y$ in the interference between different features. Although there are up to $t$ feature directions ($\boldsymbol{w}_{s=0:t-1}$) contained within the hidden state, only the feature directions that project onto $\boldsymbol{w}_y$ at timestep $t$ will affect the RNN output $\hat{y}_t$. For visual clarity in our main figures, we tie the readout $\boldsymbol{w}_y := \boldsymbol{w}_{s=k}$ (see definition of $k$ below; discussion and figures replicated without weight tying in Appendix I.2; full experimental details in Appendix I).

## 3.3 K-DELAY TASK

In order to directly control for the time span for which features remain task-relevant, we consider the $k$-delay task (Jaeger, 2002), in which the model is trained to reproduce the input sequence after a fixed delay of $k$ timesteps. In particular, the RNN is tasked with producing the target output $y_t = x_{t-k}$ at each timestep ($y_t = 0$ for $t \leq k$). Therefore $k$ acts as a control parameter that specifies how long an input feature must be maintained in the hidden state for successful task performance. We note that this task is essentially an extension of the setup in Elhage et al. (2022) to the temporal domain and that the two are identical for $k = 0$. We use a squared-error loss given by

$$\mathcal{L} = \sum_{t=1}^{T} \|y_t - \hat{y}_t\|^2 = \sum_{t=1}^{k} \|0 - \hat{y}_t\|^2 + \sum_{t=k+1}^{T} \|x_{t-k} - \hat{y}_t\|^2 \tag{4}$$

**Task-relevant and irrelevant features.** To provide some intuition about the $k$-delay task, we can refer to features in terms of their utility. At each timestep $t \geq k + 1$, there is an *output feature direction* ($\boldsymbol{w}_{s=k}$) which functions to produce $\hat{y}_t \approx x_{t-k}$ by projecting onto the readout $\boldsymbol{w}_y$. There are also *intermediate feature directions* ($\boldsymbol{w}_s$ for $0 \leq s < k$), which represent input features ($x_{t-k+1} : x_t$) from the relevant $k$-length memory window. These intermediate features are held in memory to be read out at future timesteps. Both output features and intermediate features ($k + 1$ features in total) are *task-relevant* features at timestep $t$. These features contribute to task performance either immediately or in the future, so the model is incentivized to represent them as faithfully as possible.

There also potentially exist historical, *task-irrelevant* features. These correspond to features ($x_{t-s}$ represented by $\boldsymbol{w}_s$) from the more distant ($s > k$) past. Being beyond the $k$ (shift) window, these features cannot contribute to current or future task performance. The model is therefore incentivized to forget these features (which may otherwise interfere with other task-relevant features).

## 3.4 PROJECTION AND COMPOSITION INTERFERENCE

Now that we've defined feature directions, we can understand how they interact and potentially interfere with each other. Here we introduce and define two forms of interference that occur in RNNs: projection interference and composition interference.

**Projection interference.**    Projection interference occurs when the activation of a feature is read-out as an activation of a different feature, as shown in Figure 1c. This occurs when a feature is represented ($\boldsymbol{w}_s$) non-orthogonally to the readout ($\boldsymbol{w}_y$), causing an unintended non-zero projection onto that read-out. For example, say feature direction $\boldsymbol{w}_{s=A}$ and feature direction $\boldsymbol{w}_{s=B}$ ($A \neq B$) are both non-orthogonal to the read-out $\boldsymbol{w}_y$. Then, an input ($x_{t-A}$) from timestep $t-A$ can be mistakenly read-out as having been an input ($x_{t-B}$) from timestep $t-B$, and vice versa.

**Composition interference.**    Composition interference occurs when the activation of multiple features is linearly combined into an activation that imitates another feature (Figure 1d). In the example above, if the RNN receives inputs $x_{t-A}$ and $x_{t-B}$, their feature directions ($\boldsymbol{w}_{s=A}$, $\boldsymbol{w}_{s=B}$) are co-activated and linearly combine to form $\boldsymbol{w}_{s=A} + \boldsymbol{w}_{s=B}$. A consequence is that if there is another feature direction $\boldsymbol{w}_{s=C}$ similar to this combination, the activation for inputs $x_{t-A}$ and $x_{t-B}$ will be indistinguishable from that of a single input $x_{t-C}$ from timestep $t-C$, which never occurred.

## 4    RESULTS

### 4.1    LOSS DECOMPOSITION REVEALS GEOMETRIC STRATEGY

In order to better understand how the learning problem incentivizes certain geometric strategies, we begin by studying the loss of linear RNNs. By assuming temporal independence and sparsity of the data (Appendix A), we derive (Appendix B) a form of the loss comprised of four terms.

$$
\mathbb{E}[\mathcal{L}] = \sum_{t=k+1}^{T} \left( \underbrace{p\nu \left\| \boldsymbol{w}_y^\top \boldsymbol{w}_{s=k} - 1 \right\|^2}_{\text{task benefit}} - \underbrace{2p^2\mu^2 \sum_{\substack{s \neq k}}^{t-1} \boldsymbol{w}_y^\top \boldsymbol{w}_s}_{\text{mean correction}} \right)
$$
$$
+ \sum_{t=1}^{T} \left( \underbrace{p\nu \sum_{\substack{s \neq k}}^{t-1} \left( \boldsymbol{w}_y^\top \boldsymbol{w}_s \right)^2}_{\text{projection interference cost}} + \underbrace{p^2\mu^2 \sum_{\substack{s \neq s'}}^{t-1} \left( \boldsymbol{w}_y^\top \boldsymbol{w}_s \right) \cdot \left( \boldsymbol{w}_y^\top \boldsymbol{w}_{s'} \right)}_{\text{composition interference}} \right) \tag{5}
$$

$p$ controls temporal sparsity (how frequently features occur in time; smaller $p$ corresponds to higher sparsity), and $\mu$ and $\nu$ are the mean and variance of the input distribution, respectively. By studying the terms above, we can understand the competing incentives in the loss.

First, the task benefit term is the value of successfully performing the task by aligning the feature direction $\boldsymbol{w}_{s=k}$ of the input $x_{t-k}$ to the readout $\boldsymbol{w}_y$. This corresponds to producing $x_{t-k}$ at time $t$, as required by the task. The mean correction term acts to offset any non-zero mean of the input distribution by exploiting projection interference – the RNN uses projection interference ($\boldsymbol{w}_y^\top \boldsymbol{w}_{s \neq k}$) as a bias in the absence of one and, in fact, the term disappears if we include a bias term in the output or trivially if the mean $\mu$ is 0.

Next, we can see the effects of interference in the loss. The projection interference cost introduces a penalty on feature directions $\boldsymbol{w}_s$ that project onto the readout $\boldsymbol{w}_y$ at the incorrect time ($s \neq k$). Additionally, the composition interference term comes into play when there are multiple features simultaneously active. Geometrically, this term penalizes positive correlations between $\boldsymbol{w}_s$ vectors while rewarding negative correlations, with respect to their projection onto $\boldsymbol{w}_y$, such that negative (destructive) interference is preferred over positive (constructive) interference. This essentially encourages $\boldsymbol{w}_s$ vectors to spread out in activation space as much as possible, ideally forming antipodal pairs (similar to Elhage et al. (2022) and analogous to the form in Saxe et al. (2014)).

We train linear RNNs on the $k$-delay task and see that our expected value of the loss closely predicts the empirical loss (Figure 2 *top*; Appendix C for more discussion and visualization of singular/eigenvalues and geometry throughout training). We also observe several different stages of learning, corresponding to unique changes in each of the loss terms that map onto specific geometric configurations (Figure 2 *bottom*). In particular, the RNN initially aligns all feature directions ($\boldsymbol{w}_s$) to the readout ($\boldsymbol{w}_y$) (quantified by output projection $\boldsymbol{w}_y^\top \boldsymbol{w}_s$), corresponding to a decrease in the task error term and increase in magnitude in the other three terms. After this initial alignment, feature directions begin to spread out in activation space based on their temporal ordering, causing the task error, mean correction, and composition interference terms to decrease in magnitude, while projection interference increases. The 'staircase' loss corresponds to the learning of singular values

Figure 2: **Learning dynamics characterized by initial readout alignment, then separation of feature directions.** For a 2D linear RNN on the 3-delay task. (*Top*) The expected loss $\mathbb{E}[\mathcal{L}]$ matches the empirical loss curve. The four loss terms exhibit different dynamics during training corresponding to the geometric configuration of feature directions. (*Bottom left*) Quantifying the projection onto the readout for each feature direction via the output projection $\boldsymbol{w}_y^\top \boldsymbol{w}_s$ (for $0 \le s < 12$), there is an initial readout alignment followed by temporally-ordered separation. (*Bottom right*) The final arrangement of $\boldsymbol{w}_s$ vectors converges to a spiral, matched to the output projections. The output feature $\boldsymbol{w}_{s=3}$ (*star*) has the highest projection onto $\boldsymbol{w}_y$.

(Saxe et al., 2014; Proca et al., 2025) and is indicative of saddle-to-saddle dynamics (Jacot et al., 2022) and geometric restructuring (Haputhanthri et al., 2024). The final arrangement of feature directions the RNN converges to is shown in Figure 3b. Here, in the linear case, the RNN learns to downscale and rotate features with each recurrence (such that old features spiral into the origin), implementing a 'smooth' forgetting. Indeed, in the 2D case, a spiral sink (e.g., spectral radius $\|\boldsymbol{w}_{s=0}\| < 1$) is the optimal solution for a linear RNN in the $k$-delay task (Appendices D.2 and D.3), as old features gradually fade from the hidden state.

## 4.2 FEATURES GROUP INTO INTERFERENCE-FREE SPACE

Building on the previous section, here we consider the impact of adding a nonlinearity to the readout $\boldsymbol{w}_y$ ($\sigma_y(\cdot) = \text{ReLU}(\cdot)$). The nonlinear setting makes deriving analytic solutions to the loss more challenging. We therefore approximate the expectation of the loss in the limit of high temporal sparsity (Appendix E.1), yielding

$$\mathbb{E}[\mathcal{L}] \approx p\nu \left( \underbrace{\sum_{t=k+1}^{T} (\text{ReLU}(\boldsymbol{w}_y^\top \boldsymbol{w}_{s=k}) - 1)^2}_{\text{task benefit}} + \underbrace{\sum_{t=1}^{T}\sum_{s \neq k}^{t-1} \text{ReLU}(\boldsymbol{w}_y^\top \boldsymbol{w}_s)^2}_{\text{projection interference cost}} \right) + O(p^2) \tag{6}$$

This expression resembles the one in the linear case, but the inclusion of the ReLU activation has a significant impact on its geometric interpretation. Due to the ReLU, the model only produces output for vectors that have a positive projection onto $\boldsymbol{w}_y$. Thus, all $\boldsymbol{w}_s$ vectors in the half-space opposite of $\boldsymbol{w}_y$ do not contribute to projection interference. In fact, in the extremely sparse regime (where composition interference becomes negligible), this half-space essentially becomes *interference free*. This reveals a remarkable incentive for the model to take advantage of this phenomenon by packing as many $\boldsymbol{w}_{s \neq k}$ vectors into this half-space as possible (Figure 3a).

To test this prediction, we train linear RNNs with ReLU readouts (SSMs) on the $k$-delay task at various levels of sparsity. Because the recurrence is still linear, isolation of feature directions $\boldsymbol{w}_{s \neq k}$ into this space is not always perfectly possible as $\boldsymbol{w}_s$ can only be spaced equally (along an elliptical spiral). Despite this, by looking at Figure 3c, we can see that when sparsity is high, models learn to minimize projection interference by grouping the largest feature directions into the interference-free space. The SSM employs an approximation of the strategy in which the largest feature directions

Figure 3: **Theoretical predictions match representational strategies of expressive models. (a)** A ReLU readout creates an interference-free space. In the panel, we display an idealized form, where all intermediate feature directions group into the free space (*gray shading*) and only the output feature direction lies outside of it (*star*). **(b)** Linear RNNs lack an interference-free space and instead arrange old features to spiral into the origin **(c)** In the sparse regime, SSMs minimize projection interference by grouping the largest feature directions into the interference-free space. **(d)** Nonlinear RNNs are expressive enough to fully exploit the interference-free space by grouping all of the intermediate features, separate from the output feature, and implement sharp forgetting.

$w_s$ occupy the interference-free space while smaller feature directions lie outside of it. Within the constraints of linear recurrence, this strategy still minimizes projection interference by exploiting the interference-free space. We also observe that the spectral radius (i.e., $\|w_{s=0}\|$) increases with $k$, regardless of sparsity, to overcome projection interference (Appendix E.2 for additional simulations).

### 4.3 PHASE TRANSITION FROM DENSE TO SPARSE CONFIGURATION

By varying sparsity, we can observe the existence of two discrete regimes (Figure 4). When the SSM is trained on data that is dense (low sparsity), it learns a 'dense-regime' solution where it arranges features into a spiral sink, similar to the fully-linear RNN. Although it has a nonlinearity on the readout, the SSM does not group the largest feature directions into the interference-free space in this case. We speculate that this is due to the increased likelihood of composition interference occurring when features are dense. In particular, if the SSM groups the largest feature directions into the (negative) interference-free space (e.g., $w_{s=A}$) and they additively sum with smaller (positive) feature directions ($w_{s=B}$), the ReLU will cause the model to output 0 ($\text{ReLU}(w_y^\top w_{s=A} + w_y^\top w_{s=B}) = 0$).

Instead, when inputs are sparse, it is optimal for the SSM to take advantage of the interference-free space, characterizing a 'sparse-regime' solution. The most noticeable difference in geometry between dense and sparse regimes is the angle that the task-relevant feature directions span ($k\theta$ is the angle traversed from $w_{s=0}$ to $w_{s=k}$), which exhibits a sharp change as sparsity is varied. In the dense regime, task-relevant feature directions group into a smaller cone ($\approx 90°$). Instead, in the sparse regime, task-relevant feature directions spread out into approximately $270°$ of the plane, traversing the entirety of the interference-free space to reach the readout direction. By varying sparsity, we can interpolate between these two regimes and observe a phase transition in $k\theta$, accompanied by a decrease in spectral radius ($\rho = \|w_{s=0}\|$). The difference in spectral radius is likely because in the dense regime, the SSM uses larger feature directions to compensate for projection interference (by having a large projection onto the readout that outweighs other projections). In the sparse regime, projection interference is less prevalent and therefore the SSM does not need a spectral radius as large.

Figure 4: **There exists a phase transition in optimal feature geometry between dense and sparse regimes.** As temporal sparsity increases, (*bottom*) the angular distribution of feature directions ($k\theta$) exhibits a phase transition, (*top*) accompanied by a decrease in spectral radius ($\rho$).

### 4.4 NONLINEAR RNNS EXPLOIT INTERFERENCE-FREE SPACE

We've used RNNs with linear recurrence to build our understanding of behavior in analytically tractable settings. However, these models have limited expressivity because their dynamics are constrained to a particular form, such as a spiral. Here, we consider models with nonlinear recurrence and find that, in the sparse regime, they consistently implement our predicted ideal strategy of packing as many $w_s$ feature directions into the interference-free space as possible.

We consider the nonlinear RNN model, now with both $\sigma_h(\cdot), \sigma_y(\cdot) = \text{ReLU}(\cdot)$. Although it's not possible to find a simplified expression for the feature direction $\boldsymbol{w}_s$ analytically, it still acts as the direction the input $x_{t-s}$ is represented in the limit of high sparsity ($p \to 0$; Appendix F.2), Furthermore, the interference-free space still exists and can be better exploited as the RNN can form more optimal arrangements of $\boldsymbol{w}_s$ vectors due to increased expressivity of the recurrence. In Figure 3d, we verify our predictions: nonlinear RNNs learn to pack the $k$ intermediate features into the interference-free space, with only $\boldsymbol{w}_{s=k}$ lying outside (Appendix F.3 for more simulations). Often, the vectors compress into a single quadrant. This is because the nonlinearity creates a *privileged basis* in the hidden state – positive activations are unaffected by the ReLU, while negative activations will be set to zero in the next timestep. Hence, task-relevant feature directions arrange themselves in the positive quadrant of $\boldsymbol{h}_t$.

While a RNN with linear recurrence can only implement smooth forgetting (with spectral radius $\|\boldsymbol{w}_{s=0}\| < 1$) by shrinking an input's contribution to the hidden state over time, the ReLU activation makes it possible to immediately forget a feature by sending it to the negative quadrant of $\boldsymbol{h}_t$. As a result, the nonlinear RNN can implement sharp forgetting, which enables the model to represent only task-relevant features (and remove the possibility of interference from the distant past).

### 4.5 Interaction of spatial and temporal superposition

We've focused on the case of scalar inputs and outputs in 2D space to isolate the effects of temporal superposition and for ease of visualization. We now study the interaction of spatial and temporal superposition by considering vector inputs (to introduce spatial superposition) and changing $k$ (temporal superposition). Recall that for $k = 0$, the RNN is tasked with imitating a feedforward network where it immediately outputs the input; hence, there is no need to represent features from earlier timesteps and we recover the setting of pure spatial superposition from Elhage et al. (2022).

In Figure 5, as $k$ increases, we see how a nonlinear RNN attempts to balance representing the most important features (scaling of loss) across time (A being most important, E being least). Initially, when $k = 1$, it represents features A, B, and C for 2 timesteps and drops D and E altogether. For higher $k$, the RNN eventually drops all features except A. Hence, we can see a tradeoff between the RNN's incentive to represent multiple input features and the duration of time each must be represented for. The RNN's strategy is 'all-or-none': to gain any advantage from representing a specific feature, the RNN must be able to maintain it in memory for all $k + 1$ timesteps – otherwise it will not meaningfully contribute to decreasing the loss. If the RNN does not have sufficient capacity to represent the feature for all $k + 1$ timesteps, it won't represent the feature at all. This is why we see $k + 1$ feature directions for feature A for all $k$, while other features only occur for lower $k$ when the RNN has sufficient capacity.

**Higher-dimensional hidden states.** Up to now, we have restricted the hidden state ($N_h$) of our models to 2 dimensions for easier visualization and interpretability. To extend our setting to higher-dimensions, we train nonlinear RNNs on 10-dimensional input ($N_x = 10$) on the 2-delay task, varying hidden size ($N_h = 2, 5, 10$), and measure the projection of each feature direction onto the readout ($W_y^\top W_s$). Based on our previous results, we would expect $W_y^\top W_{s=2}$ to have a diagonal of positive outputs (corresponding to the output feature directions positively projecting onto the readout: for the correct output at the correct time). Moreover, we would expect the rest of the entries in the matrix (as well as all of $W_y^\top W_{s\neq 2}$) to be negative or 0, lying in the interference-free space. Across all hidden sizes, we see this exact strategy (Figure 12 and Appendix G), with RNNs with larger hidden sizes simply capturing more features along the diagonal of $W_y^\top W_{s=2}$ (i.e., the same all-or-none effect described above). Finally, we quantify this behavior by computing the mean of the non-output feature direction projections onto the readout (i.e., $\text{mean}(W_y^\top W_{s\neq k})$) which should be negative in an optimal model, and the mean of the output feature direction projections onto the readout (i.e., $\text{mean}(\text{diag}(W_y^\top W_{s=k}))$), which should be positive. We train RNNs with hidden size 100 on a 2-delay task with 75 features and find that the best performing models group the largest feature directions into an interference-free space and project the output feature onto the readout at the appropriate time, as predicted (Figure 13).

## 5 Related Work

**Representational geometry and interpretability.** The study of representational geometry can provide better understanding of how distributed activity encodes information for different behavior. ANNs can exhibit different representational geometries, reflective of the tasks trained on

Figure 5: **There exists a tradeoff between spatial and temporal superposition.** By setting $k = 0$, we recover purely spatial superposition (as there is no memory demand) and all 5 features are arranged in a pentagon (feature legend as in Figure 1; interference-free space indicated by blue shading). As $k$ and the corresponding memory demand increases, the RNN prioritizes the most important features in an all-or-none fashion. There is a preference to represent one feature for all $k + 1$ relevant timesteps, as opposed to representing several different features for a shorter duration.

(Johnston & Fusi, 2023; 2024), dataset size (Henighan et al., 2023), architecture (Jacot et al., 2018; Chizat et al., 2019), and weight parameterization (Flesch et al., 2022; Braun et al., 2022). In addition to advancing our understanding of the solutions ANNs learn in different settings, this can also help neuroscientists identify computational structures in the brain and understand their functional roles (Saxe et al., 2020; Ostojic & Fusi, 2024). In this work, we introduce the concept of temporal superposition and study its effect on representational geometry. Our work is inspired by Elhage et al. (2022), one of the first papers to formalize superposition in toy models and study it explicitly. The field of mechanistic interpretability has since centered around the problem of superposition and developing methods to identify meaningful features from model activations (Bricken et al., 2023; Templeton et al., 2024). While these techniques have continued to advance, there has been notably less work devoted to understanding the feature geometry induced by superposition (Sharkey et al., 2025). Here we expand the study of superposition to recurrent architectures to show how memory acts as a capacity constraint, inducing superposition, and how this affects underlying geometry.

**Recurrent neural networks.** RNNs are important for modeling temporal data and studying dynamic processes. Recently, RNNs (SSMs) with linear recurrence have become popular due to their computational and memory efficiency (Gu et al., 2022b; Orvieto et al., 2023; Gu & Dao, 2024), often initialized with complex-valued parameterizations (Gu et al., 2022a; Orvieto et al., 2023), with increased expressivity (Ran-Milo et al., 2024; Orvieto et al., 2024). We find that complex eigenvalues support temporal superposition by rotating features within an interference-free space. We also study a SSM, showing that it exploits the interference-free space in the sparse regime, but is still constrained in expressivity compared to nonlinear RNNs, resulting in different geometries (and smooth vs sharp forgetting). In neuroscience, RNNs are common for modeling (Barak, 2017), as the brain's connectivity is highly recurrent and RNNs often replicate neural activity recorded in animals (Vyas et al., 2020; Khona & Fiete, 2021) and behavior (Ji-An et al., 2025) when trained on the same tasks. Similar to the feature geometry we see here, previous work in RNNs has observed rotational dynamics/sequential activity (Rajan et al., 2016; Orhan & Ma, 2019; Cueva et al., 2020; Zhang et al., 2021) for tasks with fixed delay, thought to encode temporal information. Other work has shown that RNNs trained on tasks with random delays instead exhibit persistent activity, in the form of fixed point attractors (Orhan & Ma, 2019; Liu et al., 2021; Xie et al., 2022b), similar to pure spatial superposition (e.g., when $k = 0$). Related to the interference-free space we study in our model, RNNs trained in motor-preparation paradigms similarly develop output-null subspaces where intermediate preparatory activity does not affect behavior (Schimel et al., 2024). There's been substantial theoretical work on RNNs, both by neuroscientists studying properties of neural computation and by deep learning theorists (Dubreuil et al., 2022; Driscoll et al., 2024; Schuessler et al., 2024; Zucchet & Orvieto, 2024; Proca et al., 2025). One important line of theoretical work has studied low-rank RNNs (Mastrogiuseppe & Ostojic, 2018; Schuessler et al., 2020a; Beirán et al., 2020; Dubreuil et al., 2022). These interpretable models have low-dimensional recurrent dynamics, allowing their exact phase portraits to be visualized; furthermore, these dynamics can be directly related to the underlying connectivity statistics. Related to our work, low-rank connectivity also acts as a form of capacity constraint, although the effects of such constraints have not been studied explicitly (but see Beirán et al. (2023) for comparison between low-versus full-rank RNNs). However, there is less work studying feature geometry in the context of capacity constraints induced by memory. Most work implicitly assumes an overparameterized regime (relative to task demands) when study-

ing properties of RNNs (Cohen-Karlik et al., 2023). One exception is François et al. (2025) which studied the $k$-delay task in an underparameterized linear RNN in the frequency domain. While they focused on the dense (linear) regime, here we also study the sparse regime in nonlinear RNNs, identifying novel behavior.

**Memory capacity.**   Memory capacity has previously been studied in RNNs (White et al., 2004; Dambre et al., 2012; Ballarin et al., 2024), most classically in echo-state networks (Jaeger, 2002) (we refer to short-term memory and not other forms like associative memory). Memory capacity typically refers to a temporally-dense regime. It's known that linear memory capacity is limited to the number of neurons in the hidden state $N_h$, corresponding to a single neuron per feature (orthogonality/lack of superposition). Here, we consider how RNNs handle capacity constraints, transitioning from temporally-dense to temporally-sparse regimes. We show how training with different task demands leads to different geometric solutions, aimed at compression and increased (sparse-regime) capacity. Hence, we provide a mechanistic interpretation of memory capacity under constraints.

**Working memory and serial recall.**   Our work is related to working memory in cognitive neuroscience – a cognitive function involving short-term maintenance and manipulation of information for immediate use. Similar to the $k$-delay task, working memory is often studied using serial recall, which has previously been modeled with RNNs (Botvinick & Plaut, 2006; Ganguli et al., 2008). Two existing theories of sequence working memory involve: (1) activity slots (Luck & Vogel, 1997; 2013; Xie et al., 2022a), in which there exist a set of distinct neural subspaces for different sequence items, and (2) a resource model (Alvarez & Cavanagh, 2004; Wilken & Ma, 2005; Bays & Husain, 2008), where working memory is a limited-capacity continuous resource that is shared between items (more items leads to less capacity per item). Interestingly, in the setting we consider we effectively find both (Soni & Frank, 2025). RNNs arrange features from each timestep into separate 'slots' (directions) along which these features shift through time. In the case of superposition, slots are not orthogonal (Xie et al., 2022a), but otherwise would be with sufficient capacity. Additionally, we show that the hidden space is a continuous limited-capacity resource: as memory length ($k$) or input features ($N_x$) increase, there is more demand and features are more likely to interfere. Further, we show how limited capacity leads RNNs to represent important features, while others are dropped.

## 6 Discussion

**Summary of results.**   In this work, we study the effect of time on feature representations in RNNs, introducing the concept of temporal superposition. We identify how features can interfere through either projection or composition interference, and their corresponding effects. We derive an analytical form of the loss that decomposes into four interpretable terms, which we use to explain the resulting learned geometry of RNNs. By deriving an approximation of the loss in the limit of high temporal sparsity in nonlinear RNNs, we identify the existence of an interference-free space, which RNNs exploit to minimize projection interference. By varying temporal sparsity, we see how SSMs exhibit a phase transition from an effectively linear strategy to one that uses the free space, and that this phase transition is reflected in the angular distribution of features and spectral radius. We further show that nonlinear RNNs in the sparse regime exploit the interference-free space and implement sharp forgetting. Finally, we study how spatial and temporal superposition interact as a result of different task demands and capacity constraints, and how RNNs mediate this tradeoff.

**Limitations and future work.**   We simplify our theoretical setting by assuming temporal independence of features and studying small RNNs. We also study the sparse regime: while the assumption of sparse input features appears to be reasonable (Elhage et al., 2022), the assumption of temporal sparsity may be strong. This is dependent on the task and it's an open question how the theoretical setting considered here extends more generally. Related, we study the $k$-delay task, which requires reproduction of a sequence with a delay. An important future direction will be to characterize geometry and behavior for tasks requiring manipulation of input information and varying memory demands. Finally, we note that one major assumption of the superposition hypothesis is that features are represented linearly, as directions in activation space (the linear representation hypothesis (Park et al., 2024)). A possible objection may be to what extent this work captures realistic settings in seemingly overparameterized modern-day models. Although we study a 2D case for simplicity (but see Appendix G), memory demands decrease capacity linearly with time (Jelassi et al., 2024) and consequently finite-width RNNs tasked with learning long-term dependencies will be constrained. Moreover, superposition has already been demonstrated in LLMs (Bricken et al., 2023; Templeton et al., 2024) and our study indicates that recurrence and memory will exacerbate it.

ACKNOWLEDGMENTS

AP is funded by the Imperial College London President's PhD Scholarship. LP is supported by the UKRI Centre for Doctoral Training in Safe and Trusted AI [EP/S0233356/1].

REPRODUCIBILITY STATEMENT

To ensure reproducibility of our work, we provide code to replicate all of our experiments and figures at `https://github.com/kashparty/iclr-rnn-superposition` and include experimental details in Appendix I. For our theoretical results, we include our assumptions about the data in Appendix A and full derivations and proofs in Appendices B, D.1, E.1 and F.2. We also provide additional simulations to those in the main text in Appendices C, D.2, D.3, E.2, F.3, G and H.

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

# Appendix

T ABLE OF C ONTENTS

## A  ASSUMPTIONS ABOUT DATA

We make a few simplifying assumptions about the form of the data to make our setting analytically tractable:

**Assumption 1 (temporal independence):** Inputs are generated from an IID stochastic process of scalar random variables $\{X_t\}_{t \geq 1}$.

**Assumption 2 (temporal sparsity):** Following Elhage et al. (2022), we let $X_t = B_t U_t$, where $B_t \sim \text{Bernoulli}(p)$, $U_t$ are identically distributed according to any distribution and all $\{B_t\}_{t \geq 1} \cup \{U_t\}_{t \geq 1}$ are mutually independent.

Our second assumption allows us to explicitly control temporal sparsity by varying $p$: smaller (larger) $p$ corresponds to higher (lower) sparsity. Setting $p = 1$ recovers an arbitrary IID stochastic process.

While temporal independence may be a strong assumption, it is standard in the literature on memory capacity (Jaeger, 2002; Ballarin et al., 2024). Temporal sparsity may also be a strong assumption, depending on the task or setting considered.

## B  DECOMPOSING THE $k$-DELAY LOSS

Here, we derive a simplified form of the loss of a linear RNN trained on the $k$-delay task in terms of four interpretable terms.

### B.1  DERIVATION

Under the assumptions in Appendix A, the expected value of the squared-error loss (Equation (4)) incurred by a linear RNN simplifies as follows:

$$\mathbb{E}[\mathcal{L}] = \mathbb{E}\left[\sum_{t=1}^{k}\left\|0 - \hat{Y}_t\right\|^2 + \sum_{t=k+1}^{T}\left\|X_{t-k} - \hat{Y}_t\right\|^2\right] \tag{7}$$

$$= \mathbb{E}\left[\sum_{t=1}^{k}\hat{Y}_t^2\right] + \mathbb{E}\left[\sum_{t=k+1}^{T}\left(X_{t-k} - \hat{Y}_t\right)^2\right] \tag{8}$$

$$= \sum_{t=1}^{k}\mathbb{E}\left[\hat{Y}_t^2\right] + \sum_{t=k+1}^{T}\mathbb{E}\left[\left(X_{t-k} - \hat{Y}_t\right)^2\right] \tag{9}$$

$$= \sum_{t=1}^{T}\mathbb{E}\left[\hat{Y}_t^2\right] + \sum_{t=k+1}^{T}\mathbb{E}\left[X_{t-k}^2\right] - 2\sum_{t=k+1}^{T}\mathbb{E}\left[X_{t-k}\hat{Y}_t\right] + \sum_{t=k+1}^{T}\mathbb{E}\left[\hat{Y}_t^2\right] \tag{10}$$

$$= \sum_{t=k+1}^{T}\left(\mathbb{E}\left[X_{t-k}^2\right] - 2\mathbb{E}\left[X_{t-k}\hat{Y}_t\right]\right) + \sum_{t=1}^{T}\mathbb{E}\left[\hat{Y}_t^2\right] \tag{11}$$

$$= \sum_{t=k+1}^{T}\left(\mathbb{E}\left[X_{t-k}^2\right] - 2\mathbb{E}\left[X_{t-k}\sum_{s=0}^{t-1}\boldsymbol{w}_y^\top\boldsymbol{w}_s X_{t-s}\right]\right) + \sum_{t=1}^{T}\mathbb{E}\left[\left(\sum_{s=0}^{t-1}\boldsymbol{w}_y^\top\boldsymbol{w}_s X_{t-s}\right)^2\right] \tag{12}$$

We proceed by computing each of these expectations separately. Recalling that, by the temporal sparsity assumption, $X_t = B_t U_t$, we let $\mu := \mathbb{E}[U_t]$ and $\nu := \mathbb{E}[U_t^2]$. Then, as $\{X_t\}_{t \geq 1}$ are assumed to be IID, the first expectation becomes

$$\mathbb{E}\left[X_{t-k}^2\right] = \mathbb{E}\left[B_{t-k}^2 U_{t-k}^2\right] = \mathbb{E}\left[B_{t-k}^2\right]\mathbb{E}\left[U_{t-k}^2\right] = \mathbb{E}\left[B_{t-k}\right]\mathbb{E}\left[U_{t-k}^2\right] = p\nu \tag{13}$$

Next, in computing the second expectation, we must handle the case of $s = k$ separately. This yields

$$\mathbb{E}\left[X_{t-k}\sum_{s=0}^{t-1}\boldsymbol{w}_y^\top\boldsymbol{w}_s X_{t-s}\right] = \mathbb{E}\left[X_{t-k}^2\right]\boldsymbol{w}_y^\top\boldsymbol{w}_{s=k} + \sum_{\substack{s=0\\s\neq k}}^{t-1}\mathbb{E}\left[X_{t-k}X_{t-s}\right]\boldsymbol{w}_y^\top\boldsymbol{w}_s \tag{14}$$

$$= p\nu\boldsymbol{w}_y^\top\boldsymbol{w}_{s=k} + \sum_{\substack{s=0\\s\neq k}}^{t-1}\mathbb{E}\left[B_{t-k}U_{t-k}B_{t-s}U_{t-s}\right]\boldsymbol{w}_y^\top\boldsymbol{w}_s \tag{15}$$

$$= p\nu\boldsymbol{w}_y^\top\boldsymbol{w}_{s=k} + p^2\mu^2\sum_{\substack{s=0\\s\neq k}}^{t-1}\boldsymbol{w}_y^\top\boldsymbol{w}_s \tag{16}$$

For the third expectation, we must handle the diagonal terms (where $s = s'$) separately:

$$\mathbb{E}\left[\left(\sum_{s=0}^{t-1}\boldsymbol{w}_y^\top\boldsymbol{w}_s X_{t-s}\right)^2\right] = \mathbb{E}\left[\sum_{s=0}^{t-1}\left(\boldsymbol{w}_y^\top\boldsymbol{w}_s\right)^2 X_{t-s}^2 + \sum_{\substack{s,s'=0\\s\neq s'}}^{t-1}\left(\boldsymbol{w}_y^\top\boldsymbol{w}_s\right)\left(\boldsymbol{w}_y^\top\boldsymbol{w}_{s'}\right)X_{t-s}X_{t-s'}\right] \tag{17}$$

$$= \sum_{s=0}^{t-1}\left(\boldsymbol{w}_y^\top\boldsymbol{w}_s\right)^2\mathbb{E}\left[X_{t-s}^2\right] + \sum_{\substack{s,s'=0\\s\neq s'}}^{t-1}\left(\boldsymbol{w}_y^\top\boldsymbol{w}_s\right)\left(\boldsymbol{w}_y^\top\boldsymbol{w}_{s'}\right)\mathbb{E}\left[X_{t-s}X_{t-s'}\right] \tag{18}$$

$$= p\nu\sum_{s=0}^{t-1}\left(\boldsymbol{w}_y^\top\boldsymbol{w}_s\right)^2 + p^2\mu^2\sum_{\substack{s,s'=0\\s\neq s'}}^{t-1}\left(\boldsymbol{w}_y^\top\boldsymbol{w}_s\right)\left(\boldsymbol{w}_y^\top\boldsymbol{w}_{s'}\right) \tag{19}$$

Substituting these expressions into equation 12, we obtain

$$\mathbb{E}[\mathcal{L}] = \sum_{t=k+1}^{T}\left(p\nu - 2p\nu\boldsymbol{w}_y^\top\boldsymbol{w}_{s=k} - 2p^2\mu^2\sum_{\substack{s=0\\s\neq k}}^{t-1}\boldsymbol{w}_y^\top\boldsymbol{w}_s\right) \tag{20}$$

$$+ \sum_{t=1}^{T}\left(p\nu\sum_{s=0}^{t-1}\left(\boldsymbol{w}_y^\top\boldsymbol{w}_s\right)^2 + p^2\mu^2\sum_{\substack{s,s'=0\\s\neq s'}}^{t-1}\left(\boldsymbol{w}_y^\top\boldsymbol{w}_s\right)\left(\boldsymbol{w}_y^\top\boldsymbol{w}_{s'}\right)\right) \tag{21}$$

Finally, we move every occurrence of the $p\nu\left(\boldsymbol{w}_y^\top\boldsymbol{w}_{s=k}^2\right)^2$ term into the first summation:

$$\mathbb{E}[\mathcal{L}] = \sum_{t=k+1}^{T}\left(p\nu - 2p\nu\boldsymbol{w}_y^\top\boldsymbol{w}_{s=k} + p\nu\left(\boldsymbol{w}_y^\top\boldsymbol{w}_{s=k}\right)^2 - 2p^2\mu^2\sum_{\substack{s=0\\s\neq k}}^{t-1}\boldsymbol{w}_y^\top\boldsymbol{w}_s\right) \tag{22}$$

$$+ \sum_{t=1}^{T}\left(p\nu\sum_{\substack{s=0\\s\neq k}}^{t-1}\left(\boldsymbol{w}_y^\top\boldsymbol{w}_s\right)^2 + p^2\mu^2\sum_{\substack{s,s'=0\\s\neq s'}}^{t-1}\left(\boldsymbol{w}_y^\top\boldsymbol{w}_s\right)\left(\boldsymbol{w}_y^\top\boldsymbol{w}_{s'}\right)\right) \tag{23}$$

After factorizing, we arrive at the form:

$$\mathbb{E}[\mathcal{L}] = \sum_{t=k+1}^{T} \left( \underbrace{p\nu \left\| \boldsymbol{w}_y^\top \boldsymbol{w}_{s=k} - 1 \right\|^2}_{\text{task benefit}} - \underbrace{2p^2\mu^2 \sum_{\substack{s=0 \\ s\neq k}}^{t-1} \boldsymbol{w}_y^\top \boldsymbol{w}_s}_{\text{mean correction}} \right) \tag{24}$$

$$+ \sum_{t=1}^{T} \left( \underbrace{p\nu \sum_{\substack{s=0 \\ s\neq k}}^{t-1} \left( \boldsymbol{w}_y^\top \boldsymbol{w}_s \right)^2}_{\text{projection interference cost}} + \underbrace{p^2\mu^2 \sum_{\substack{s,s'=0 \\ s\neq s'}}^{t-1} \left( \boldsymbol{w}_y^\top \boldsymbol{w}_s \right) \cdot \left( \boldsymbol{w}_y^\top \boldsymbol{w}_{s'} \right)}_{\text{composition interference}} \right) \tag{25}$$

We omit the 0 indexing of summations in the main text to reduce clutter.

### B.2 DISCUSSION ON MEAN CORRECTION AND COMPOSITION INTERFERENCE TERMS

By looking at expectation of the loss, we can see that the mean correction and composition interference terms only exist when the data distribution has non-zero mean. However, we can also see that the two terms are scaled by $p^2$. This implies that, even for data with a non-zero mean, in the sparse regime where we take the limit of $p \to 0$, the impact of these two terms on the loss becomes negligible. The mean correction term disappears because, even though the underlying data distribution $U$ has a fixed mean $\mathbb{E}[U] = \mu$, the mean of the overall input $X_t$ is proportional to its sparsity: $\mathbb{E}[X] = \mathbb{E}[B]\mathbb{E}[U] = p\mu$, so $\mathbb{E}[X] \to 0$ as $p \to 0$ and sparse data does have an approximately zero mean. Furthermore, the composition interference term disappears because the probability of two or more temporal features simultaneously activating is $O(p^2)$, so composition interference becomes negligible as $p \to 0$.

## C GEOMETRIC INTERPRETATION OF LEARNING DYNAMICS

Building on earlier work in feedforward linear networks (Saxe et al., 2014), recent work has investigated the learning dynamics of RNNs (Schuessler et al., 2020b; Proca et al., 2025; van Rossem & Saxe, 2025; Ger & Barak, 2025; Bordelon et al., 2025). It's been shown that neural networks initialized with small random weights appear to undergo an initial phase of eigenvector alignment (Atanasov et al., 2022), followed by learning of the largest (and latest occurring) data correlation singular values/eigenvalues (Saxe et al., 2014; Proca et al., 2025).

Having derived an exact expression of the expected loss in terms of four interpretable terms, here we study their learning dynamics to better characterize the geometric evolution of the network (Figure 2). Additionally, we visualize the dynamics of the output projection ($\boldsymbol{w}_y^\top \boldsymbol{w}_s$ for $0 \leq s < 12$) to better understand the functional behavior of the network throughout training. Taken together, this analysis provides a direct connection between the model's functionality/performance and its representational geometry.

First, we observe that our predicted expectation of the loss (in red) closely matches the empirical loss (light purple), corroborating our theoretical result. Interestingly, we see that the loss decreases in a 'staircase,' consistent with Saxe et al. (2014); Proca et al. (2025). In particular, Saxe et al. (2014) identified that in feedforward networks, the largest data correlation singular values are learned first, corresponding to staircase-like drops in the loss, while Proca et al. (2025) extended this work to RNNs and identified that task dynamics with singular/eigen-values that are large and occur later in the trajectory are learned faster (i.e., stronger correlation with the most recent past). Work in feedforward networks has linked these stage-like learning curves to trajectories passing near saddle points (saddle-to-saddle dynamics) (Jacot et al., 2022). In RNNs, a similar phenomenon has also been studied, linking abrupt learning to bifurcations caused by geometric restructuring (Haputhanthri et al., 2024). Interestingly, our results indicate that such geometric restructuring can occur without the emergence of attractors, as our model instead learns a spiral.

Figure 6: **Learning dynamics of eigenvalues, singular values, and geometry of a linear RNN. (a)** The eigenvalues of $W_h$ first learn the real part, corresponding to the alignment of feature directions $\boldsymbol{w}_s$ to the readout $\boldsymbol{w}_y$ (aligned to y-axis), followed by the learning of the complex part which forms the rotation of feature directions. **(b)** The learning of the singular values of $W_h$ approximately correspond to the staircase-like drops in the loss in Figure 2 and roughly to the learning of real and complex parts of the eigenvalues.

In Figure 6a, we visualize the eigenvalues of $W_h$ along with the corresponding geometry of the feature direction vectors $\boldsymbol{w}_s$ at different points in training. Interestingly, we see several stages of learning emerge, whereby the real part of the eigenvalues are learned first, corresponding to the alignment of feature directions with the readout. Around step 14700, the RNN seems to have learned a shear transformation. Next, the complex part is learned, causing feature directions to arrange into a spiral. Interestingly, these dynamics differ from Proca et al. (2025) in which the real and complex parts of the eigenvalues are learned simultaneously when eigenvectors are aligned with the data correlations. The staircase loss we observe in Figure 2 does not seem to match to the learning of individual eigenvalues, but it does seem to approximately match the learning of the singular values in Figure 6b (which roughly corresponds to the learning of the real then complex parts of the eigenvalues). This observation highlights an interesting relationship between the learning dynamics of singular values and (complex-valued) eigenvalues, and as such the emergence of geometric structure. An important difference to note is that Proca et al. (2025) considers linear RNNs that are not capacity-constrained (do not exhibit superposition) and have data-aligned weights. It's possible that different learning dynamics emerge for capacity-constrained RNNs/solutions relying on superposition.

## D   ANALYZING THE SPECTRAL RADIUS OF $W_h$

### D.1   PROVING THE ECHO STATE PROPERTY

In this section, we prove that in our setting, achieving finite loss in the infinite time limit ($T \to \infty$) is only possible in models with a spectral radius of $W_h$ less than 1 (the so-called echo state property), to demonstrate the optimality of certain solutions found by the networks we study. While there may exist other solutions that perform well on short sequences with a spectral radius greater or equal to 1, we show that their performance will degrade as sequence length increases.

In particular, we consider models that satisfy

$$\lim_{T \to \infty} \frac{1}{T} \mathbb{E}[\mathcal{L}(T)] < \infty \tag{26}$$

where $T$ refers to the sequence length and $\mathcal{L}(T)$ is defined as per Equation (4). We prove that this condition is only possible if the spectral radius of $W_h$ is $\rho(W_h) < 1$.

*Proof.* Consider the expected loss over an entire sequence, $\mathbb{E}[\mathcal{L}(T)]$, as a sum of terms,

$$\mathbb{E}[\mathcal{L}(T)] = L_{t=1} + L_{t=2} + \cdots + L_{t=T} \tag{27}$$

where $L_t$ is the loss incurred by the model on timestep $t$. Then, for $t > k$, we have

$$L_{t>k} = p\nu \left\| \boldsymbol{w}_y^\top \boldsymbol{w}_{s=k} - 1 \right\|^2 - 2p^2\mu^2 \sum_{\substack{s=0 \\ s \neq k}}^{t-1} \boldsymbol{w}_y^\top \boldsymbol{w}_s \tag{28}$$

$$+ p\nu \sum_{\substack{s=0 \\ s\neq k}}^{t-1} \left(\boldsymbol{w}_y^\top \boldsymbol{w}_s\right)^2 + p^2\mu^2 \sum_{\substack{s,s'=0 \\ s\neq s'}}^{t-1} \left(\boldsymbol{w}_y^\top \boldsymbol{w}_s\right) \cdot \left(\boldsymbol{w}_y^\top \boldsymbol{w}_{s'}\right) \tag{29}$$

This rearranges to

$$L_{t>k} = (p\nu - p^2\mu^2) \left( \left\| \boldsymbol{w}_y^\top \boldsymbol{w}_{s=k} - 1 \right\|^2 + \sum_{\substack{s=0 \\ s\neq k}}^{t-1} \left(\boldsymbol{w}_y^\top \boldsymbol{w}_s\right)^2 \right) + p^2\mu^2 \left( \sum_{s=0}^{t-1} \boldsymbol{w}_y^\top \boldsymbol{w}_s - 1 \right)^2 \tag{30}$$

whose terms are all non-negative as $p\nu - p^2\mu^2 = \mathrm{Var}[X_t] \geq 0$.

If the loss incurred at large enough timesteps, $L_{t>k}$, diverges to $\infty$ as $t \to \infty$, then the average loss per timestep, $\frac{1}{T}\mathbb{E}[\mathcal{L}(T)]$, also diverges to infinity. Hence, for the average loss per timestep to remain finite, we require $L_{t>k}$ to remain finite.

Now, assume that $\rho(W_h) \geq 1$ and that $L_{t>k}$ does not diverge to infinity. As all terms in the expression for $L_{t>k}$ are non-negative, the series $\sum_{s\neq k}^{\infty} \left(\boldsymbol{w}_y^\top \boldsymbol{w}_s\right)^2$ must be convergent. This implies $\lim_{s\to\infty} \boldsymbol{w}_y^\top \boldsymbol{w}_s = 0$. Expanding, we obtain

$$\lim_{s\to\infty} \boldsymbol{w}_y^\top \boldsymbol{w}_s = \lim_{s\to\infty} \boldsymbol{w}_y^\top W_h^s \boldsymbol{w}_x = \boldsymbol{w}_y^\top \left( \lim_{s\to\infty} W_h^s \right) \boldsymbol{w}_x = \boldsymbol{w}_y^\top W_h^\infty \boldsymbol{w}_x = 0 \tag{31}$$

where we denote $W_h^\infty = \lim_{s\to\infty} W_h^s$, which is non-zero as $\rho(W_h) \geq 1$. The equality holds precisely when

$$\boldsymbol{w}_x \in \ker(W_h^\infty) \quad \text{or} \quad \boldsymbol{w}_y \in \ker\left(W_h^{\infty\top}\right) \quad \text{or} \quad \boldsymbol{w}_y \perp W_h^\infty \boldsymbol{w}_x$$

Since $W_h^\infty \neq 0$, we have $\mathrm{rank}(W_h^\infty) = \mathrm{rank}(W_h^{\infty\top}) > 0$. Hence, by the rank-nullity theorem, $\dim(\ker(W_h^\infty)) = N_h - \mathrm{rank}(W_h^\infty) < N_h$ and $\dim(\ker(W_h^{\infty\top})) = N_h - \mathrm{rank}(W_h^{\infty\top}) < N_h$. Thus both $\ker(W_h^\infty)$ and $\ker(W_h^{\infty\top})$ have Lebesgue measure zero.

If $\boldsymbol{w}_x \notin \ker(W_h^\infty)$, we require the third case, where $W_y^\top$ must lie on the $(N_h - 1)$-dimensional hyperplane orthogonal to $W_h^\infty \boldsymbol{w}_x$. This is again a proper subspace of $\mathbb{R}^{N_h}$ with measure zero. Hence, if $\rho(W_h) \geq 1$, then the set of solutions for which $L_{t>k}$ remains finite has measure zero, so $L_{t>k}$ almost surely diverges to infinity. Taking the contrapositive, if $L_{t>k}$ remains finite, then $\rho(W_h) < 1$ almost surely. □

Intuitively, we have shown that if $\rho(W_h) \geq 1$, then $L_{t>k}$ diverges to infinity except if $\boldsymbol{w}_x$, $W_h$ and $\boldsymbol{w}_y$ *precisely* (not approximately) satisfy certain conditions. We can be confident that these conditions are not satisfied by models in practice: it would require the optimizer to balance the model parameters on an infinitely thin "knife edge", which is practically impossible in floating-point arithmetic. Hence we can safely restrict our attention to models satisfying $\rho(W_h) < 1$.

### D.2 Optimality of spiral sinks in 2D

In the proof above, we have shown that $\rho(W_h) < 1$ in order to have finite loss as sequence length $T \to \infty$. This condition on the spectral radius is known as the echo state property and can be interpreted as the model forgetting old inputs over time. This constraint guarantees the shrinking aspect of a spiral sink. We now verify that the spiral behavior, specifically, is the optimal solution in 2 dimensions. To do this, we train linear RNNs parameterized such that the trace and determinant of $W_h \in \mathbb{R}^{2\times 2}$ is fixed. Specifically, for each desired trace-determinant pair $(\tau, \delta)$, we optimize over the 2-dimensional manifold

$$\mathcal{M} = \left\{ W_h \in \mathbb{R}^{2\times 2} : \mathrm{tr}(W_h) = \tau, \det(W_h) = \delta \right\} \tag{32}$$

which we parameterize by $(\theta_1, \theta_2) \in \mathbb{R}^2$ using the map

$$\varphi(\theta_1, \theta_2) = \begin{bmatrix} \theta_1 & \frac{\theta_1(\tau - \theta_1) - \delta}{\exp(\theta_2)} \\ \exp(\theta_2) & \tau - \theta_1 \end{bmatrix}$$

Figure 7: **Trace-determinant classification of 2-dimensional discrete linear dynamical systems**. Adapted from Galor (2007). Stable systems occupy the triangular region enclosed by the lines $\delta = 1$, $\delta = x - 1$ and $\delta = -x - 1$. Within this triangle, spiral sinks are found above the parabola $\delta = \frac{1}{4}\tau^2$.

where $\theta_2$ is exponentiated to ensure $\varphi$ is bijective. We verify that

$$\mathrm{tr}(\varphi(\theta_1, \theta_2)) = \theta_1 + \tau - \theta_1 = \tau \tag{33}$$

$$\det(\varphi(\theta_1, \theta_2)) = \theta_1(\tau - \theta_1) - \frac{\theta_1(\tau - \theta_1)}{\exp(\theta_2)} \exp(\theta_2) = \delta \tag{34}$$

We sweep through a grid of points in the square $(\tau, \delta) \in [-2, 2]^2$ and for each point, we train a linear RNN parameterized as above. The loss achieved by each model under various task conditions is shown in Figure 8. Each plot can be thought of as a trace-determinant slice of the $k$-delay loss landscape.

We note that in every case, the optimal solution (brightest point on the plot) is found in the region that corresponds to spiral sinks. This provides strong empirical evidence that the globally optimal solution must be a spiral sink. This makes intuitive sense: rotation is used to implement an approximate delay-line solution by shifting inputs through a sequence of positions in the hidden state, while the gradual shrinking of vectors facilitates the forgetting of old inputs.

### D.3 OPTIMALITY OF ACUTE-ANGLED SPIRAL SINKS

Interestingly, the plots become increasingly symmetric with increasing sparsity. In the low-sparsity case for both the 2-delay and 4-delay tasks, the global optimum clearly has $\mathrm{tr}(W_h) > 0$, which implies an acute rotation angle. This can be seen by writing the eigenvalues of $W_h$ as

$$\lambda_{1,2} = \frac{t}{2} \pm \frac{i}{2}\sqrt{4\delta - \tau^2} \tag{35}$$

and observing that $t < 0$ corresponds to $\arg(\lambda_{1,2}) \in \left(\frac{\pi}{2}, \frac{3\pi}{2}\right)$ (obtuse rotation angle) while $t > 0$ corresponds to $\arg(\lambda_{1,2}) \in \left(-\frac{\pi}{2}, \frac{\pi}{2}\right)$ (acute rotation angle).

The asymmetry is most clearly visible in the figure for the 2-delay task at 0.7 sparsity. In Figure 9, we decompose this particular loss landscape to investigate the reason for acute-angled rotation being preferred. While none of the terms are perfectly symmetric in $\mathrm{tr}(W_h)$, it is clear that the mean correction term is largely driving this behavior: for acute spirals, it can reduce loss, while for most instances of obtuse spirals, it increases the loss. Unsurprisingly, there exists a trade-off between this term and the others (in particular, composition interference largely seems positive where mean correction is negative, and vice versa), but evidently the optimal balance is firmly in the acute spiral region. Overall, there seems to be a region in which task error, mean correction and composition interference are all relatively low, while projection interference is relatively high – this corresponds precisely with the lowest-loss region in Figure 8 and is exactly the sacrifice we observe in Figure 2.

Figure 8: **Loss landscape of the 2-delay and 4-delay tasks.** Shown in terms of $\mathrm{tr}(W_h)$ and $\det(W_h)$ at 3 sparsity levels for linear RNNs with 2-dimensional $W_h$. At each point on the trace-determinant plane a linear model with 2-dimensional hidden state was parameterized, as described above, with a fixed trace and determinant. The final training loss is displayed as a multiple of (i.e. relative to) the lowest training loss achieved by any of the models. The best-performing model for each task is marked by a cross. Standard lines and curves used to classify discrete dynamical systems are overlaid in white; refer to Figure 7 for interpretation.

Figure 9: **Decomposed loss landscape of the 2-delay task in the trace-determinant plane.** Trace-determinant loss landscape of the 2-delay task at 0.7 sparsity for stable linear RNNs with 2-dimensional $W_h$, decomposed into interpretable terms. The best-performing model (in terms of overall empirical loss) is marked with a cross and annotated with its value for each of the loss terms.

# E    STATE SPACE MODEL (LINEAR RECURRENCE, NONLINEAR READOUT)

## E.1    APPROXIMATION OF LOSS UNDER HIGH SPARSITY

We now consider the SSM that produces its output through a ReLU activation function, corresponding to $\sigma_y = \text{ReLU}$ and $\sigma_h = \text{id}$. Due to the ReLU activation, such a model can only produce non-negative outputs, so for the $k$-delay task, it is sensible to restrict the input distribution to be non-negative as well (as we desire $\hat{y}_t = \text{ReLU}(\dots) = x_{t-k}$). Recall that the input distribution is $X_t = B_t U_t$, where $B_t \sim \text{Bernoulli}(p)$ and $U_t$ is are identically distributed according to any distribution. Then, since $B_t \geq 0$, we require $U_t \geq 0$ to ensure $X_t \geq 0$. For example, the distribution used by Elhage et al. (2022), where $U_t \sim \text{Uniform}[0, 1]$, would satisfy this requirement.

We now prove some key results about the expected value of the ReLU function applied to functions of a random variable $Z \geq 0$. Firstly, we note that

$$\mathbb{E}[\text{ReLU}(Z)] = \mathbb{E}[Z] \tag{36}$$

We now consider $\mathbb{E}[\text{ReLU}(VZ)]$ for an arbitrary random variable $V \in \mathbb{R}$, which is permitted to be negative. Since $Z \geq 0$, we have

$$\mathbb{E}[\text{ReLU}(VZ)] = \mathbb{E}[\text{ReLU}(V)Z] = \mathbb{E}[\text{ReLU}(V)]\mathbb{E}[Z] \tag{37}$$

In the special case of a constant $V = v$, this simplifies to $\mathbb{E}[\text{ReLU}(vZ)] = \text{ReLU}(v)\mathbb{E}[Z]$.

Now, recalling that $B_t, U_t, X_t \geq 0$ with $\mathbb{E}[B_t] = p$, $\mathbb{E}[U_t] = \mu$ and $\mathbb{E}[X_t] = p\mu$, we apply these results to the expected model output given by

$$\mathbb{E}[Y_t] = \mathbb{E}\left[\text{ReLU}\left(\sum_{s=0}^{t-1} \boldsymbol{w}_y^\top \boldsymbol{w}_s X_{t-s}\right)\right] \tag{38}$$

Previously, we were able to apply linearity of expectation to split the expectation of a sum into a sum of expectations, but the ReLU non-linearity precludes using the same approach here. The distribution of the interior weighted sum of uniformly distributed variables is known (Bradley & Gupta, 2002), but its complexity explodes with increasing $t$. We therefore opt for an approximation.

Let $\rho$ be the spectral radius of $W_h$. Then the feature $x_{t-s}$ is represented by the vector $\boldsymbol{w}_s = W_h^s \boldsymbol{w}_x$ in the hidden state $\boldsymbol{h}_t$. For old features, corresponding to large $s$, we have $\|W_h^s\| \approx \rho^s$ by Gelfand's

formula. As argued in Appendix D.1, we are only concerned with the case of $\rho < 1$, so for any $\varepsilon > 0$, there exists a "memory window" of length $T_\varepsilon = \lceil \log(\varepsilon)/\log(\rho) \rceil$ such that for $s \geq T_\varepsilon$, $\|W_h^s\| \approx \rho^s \leq \rho^{T_\varepsilon} \leq \varepsilon$. Hence the contribution of any input older than $T_\varepsilon$ has magnitude of order $O(\varepsilon)$. Intuitively, this means that if we set $\varepsilon$ small enough (and thus $T_\varepsilon$ large enough), we can ignore inputs older than $T_\varepsilon$ time steps.

Therefore, the only inputs that can have a significant effect on the model's behavior are those which arrived in the last $T_\varepsilon$ time steps. Since each input is masked by a Bernoulli random variable, the number of non-zero inputs that arrive in $T_\varepsilon$ time steps is distributed according to $N_\varepsilon \sim \mathrm{Binomial}(T_\varepsilon, p)$. This quantity essentially counts the number of inputs actually "in play," meaning that their effect on the hidden state has magnitude larger than $\varepsilon$. Hence, the probability that there are two or more such inputs is given by

$$\Pr[N_\varepsilon \geq 2] = 1 - (1-p)^{T_\varepsilon} - pT_\varepsilon (1-p)^{T_\varepsilon - 1} \approx \frac{p^2}{2} T_\varepsilon (T_\varepsilon - 1), \tag{39}$$

where the binomial approximation holds for small $p$. Therefore, if we are willing to ignore cases that arise with probability less than some $\delta > 0$, we can approximate the behavior of an RNN by its behavior on input sequences with only one non-zero input for $\Pr[N_\varepsilon \geq 2] < \delta$. This occurs when sparsity is high enough to make it vanishingly rare for two or more inputs to be "in play" simultaneously. Specifically, the approximation is valid when

$$p < \sqrt{\frac{2\delta}{T_\varepsilon (T_\varepsilon - 1)}} < \frac{\sqrt{2\delta}}{\log(1/\varepsilon)} \log(1/\rho), \tag{40}$$

or, equivalently,

$$\rho < \exp\left(-\frac{p\log(1/\varepsilon)}{\sqrt{2\delta}}\right) = \exp\left(\frac{p\log(\varepsilon)}{\sqrt{2\delta}}\right) = \varepsilon^{p/\sqrt{2\delta}}. \tag{41}$$

In particular, for arbitrarily tight $\delta, \varepsilon > 0$, there always exists a $p$ small enough to make the approximation valid for any given model with $\rho < 1$. Under this approximation, our analysis of the ReLU-gated model becomes tractable, as we can ignore all cases that involve two or more non-zero inputs. For instance, the expected model output becomes

$$\mathbb{E}\left[\mathrm{ReLU}\left(\sum_{s=0}^{t-1} \boldsymbol{w}_y^\top \boldsymbol{w}_s X_{t-s}\right)\right] \approx \sum_{s=0}^{t-1} \mathbb{E}\left[\mathrm{ReLU}\left(\boldsymbol{w}_y^\top \boldsymbol{w}_s X_{t-s}\right)\right] \tag{42}$$

$$= \sum_{s=0}^{t-1} \mathbb{E}[X_{t-s}] \, \mathrm{ReLU}\left(\boldsymbol{w}_y^\top \boldsymbol{w}_s\right) \tag{43}$$

$$= p\mu \sum_{s=0}^{t-1} \mathrm{ReLU}\left(\boldsymbol{w}_y^\top \boldsymbol{w}_s\right). \tag{44}$$

We now follow the steps of Appendix B.1, applying this assumption to derive an interpretable expression for the expected squared-error loss $\mathbb{E}[\mathcal{L}]$. We begin with

$$\mathbb{E}[\mathcal{L}] = \sum_{t=k+1}^{T} \left(\mathbb{E}\left[X_{t-k}^2\right] - 2\mathbb{E}\left[X_{t-k} \, \mathrm{ReLU}\left(\sum_{s=0}^{t-1} \boldsymbol{w}_y^\top \boldsymbol{w}_s X_{t-s}\right)\right]\right) \tag{45}$$

$$+ \sum_{t=1}^{T} \mathbb{E}\left[\mathrm{ReLU}\left(\sum_{s=0}^{t-1} \boldsymbol{w}_y^\top \boldsymbol{w}_s X_{t-s}\right)^2\right] \tag{46}$$

As before, the first term is $\mathbb{E}\left[X_{t-k}^2\right] = p\nu$. To evaluate the second term, we again use the assumption that no more than one of the inputs is non-zero. There are two cases: either $X_{t-k}$ is zero, in which case the entire term collapses to zero, or $X_{t-k}$ is non-zero, in which case all other $X_{t-s}$ are zero for $s \neq k$. Hence the second expectation simplifies to

$$\mathbb{E}\left[X_{t-k} \, \mathrm{ReLU}\left(\sum_{s=0}^{t-1} \boldsymbol{w}_y^\top \boldsymbol{w}_s X_{t-s}\right)\right] \approx \mathbb{E}\left[X_{t-k} \, \mathrm{ReLU}(\boldsymbol{w}_y^\top \boldsymbol{w}_{s=k} X_{t-k})\right] \tag{47}$$

$$= \mathbb{E}\left[X_{t-k}^2\right] \mathrm{ReLU}(\boldsymbol{w}_y^\top \boldsymbol{w}_{s=k}) \tag{48}$$

$$= p\nu \, \mathrm{ReLU}(\boldsymbol{w}_y^\top \boldsymbol{w}_{s=k}) \tag{49}$$

Similarly, in the third expectation, the only non-zero summands are those on the "diagonal" (all off-diagonal terms require two inputs to be non-zero, so we ignore them):

$$\mathbb{E}\left[\mathrm{ReLU}\left(\sum_{s=0}^{t-1} \boldsymbol{w}_y^\top \boldsymbol{w}_s X_{t-s}\right)^2\right] \approx \sum_{s=0}^{t-1} \mathbb{E}\left[\mathrm{ReLU}\left(\boldsymbol{w}_y^\top \boldsymbol{w}_s X_{t-s}\right)^2\right] \tag{50}$$

$$= \sum_{s=0}^{t-1} \mathbb{E}\left[X_{t-s}^2\right] \mathrm{ReLU}(\boldsymbol{w}_y^\top \boldsymbol{w}_s)^2 \tag{51}$$

$$= p\nu \sum_{s=0}^{t-1} \mathrm{ReLU}(\boldsymbol{w}_y^\top \boldsymbol{w}_s)^2 \tag{52}$$

Putting these together:

$$\mathbb{E}[\mathcal{L}] \approx \sum_{t=k+1}^{T}\left(p\nu - 2p\nu\,\mathrm{ReLU}(\boldsymbol{w}_y^\top \boldsymbol{w}_{s=k})\right) + \sum_{t=1}^{T} p\nu \sum_{s=0}^{t-1} \mathrm{ReLU}(\boldsymbol{w}_y^\top \boldsymbol{w}_s)^2 \tag{53}$$

$$= p\nu\left(\sum_{t=k+1}^{T}\left(\mathrm{ReLU}(\boldsymbol{w}_y^\top \boldsymbol{w}_{s=k}) - 1\right)^2 + \sum_{t=1}^{T} \sum_{\substack{s=0\\s\neq k}}^{t-1} \mathrm{ReLU}(\boldsymbol{w}_y^\top \boldsymbol{w}_s)^2\right) \tag{54}$$

### E.2 SIMULATIONS ACROSS $k$ AND SPARSITY FOR NONLINEAR READOUT

In Figure 10 we provide additional simulations of the solutions learned by optimal SSMs for different values of $k$ and sparsity $1 - p$, accompanied by the spectral radius $\rho$ and the angular distribution of task-relevant features $k\theta$.

In the SSM, we observe that the spectral radius (i.e., $\|\boldsymbol{w}_{s=0}\|$) increases with $k$, regardless of sparsity (Figure 10). This is simply because for larger $k$, the model must hold inputs in its memory for more timesteps. If the spectral radius is too small, then the magnitude of $\boldsymbol{w}_{s=k}$ in the hidden state will be negligible relative to other $\boldsymbol{w}_{s\neq k}$ and so any task-relevant signal will be overpowered by projection interference from other features.

Figure 10: **Solutions to $k$-delay learned by linear models with ReLU read-out (SSMs).** Rows correspond to different values of $k$ and columns correspond to different sparsity levels. The $\boldsymbol{w}_s$ are plotted after applying a conformal linear transformation such that the $y$-component of each $\boldsymbol{w}_s$ is $\boldsymbol{w}_y^\top \boldsymbol{w}_s$, and $\boldsymbol{w}_x = \boldsymbol{w}_{s=0}$ points towards positive $x$. Thus the interference-free half-space is simply given by $y < 0$. As before, the output feature, $\boldsymbol{w}_{s=k}$, is marked with a star. Note that the plots vary significantly in scale, so it is not meaningful to compare the magnitude of a particular $\boldsymbol{w}_s$ vector between different plots. The angle $\theta$ is calculated as $\arg(\lambda_1)$ where $\lambda_1 \in \mathbb{C}$ is an eigenvalue of $W_h$.

## F NONLINEAR RNN (NONLINEAR RECURRENCE, NONLINEAR READOUT)

### F.1 DISCUSSION ON APPLYING NONLINEARITY TO HIDDEN LAYER

Recall that we have defined the RNN architecture as

$$\boldsymbol{h}_t = W_x \boldsymbol{x}_t + W_h \sigma_h(\boldsymbol{h}_{t-1}) \qquad\qquad \hat{\boldsymbol{y}}_t = \sigma_y(W_y^\top \boldsymbol{h}_t) \qquad\qquad (55)$$

While this is a slight departure from traditional RNN architecture in machine learning (with regards to what we label as $\boldsymbol{h}_t$ and how the output $\hat{\boldsymbol{y}}_t$ is computed), this form is standard in computational neuroscience. Moreover, prior work has shown that the recurrence in these models is mathematically equivalent (Miller & Fumarola, 2012).

$$\text{ML } \boldsymbol{h}_t = \sigma_h \left( \underbrace{W_x \boldsymbol{x}_t + W_h \boldsymbol{h}_{t-1}}_{\text{our/neuro } \boldsymbol{h}_t} \right) \qquad\qquad (56)$$

The motivation for using this form of RNN is that it allows us to study a nonlinear RNN for which the linear representation hypothesis provably holds (under certain sparsity conditions), as shown in Appendix F.2, and is the most direct extension of Elhage et al. (2022) to the recurrent setting.

### F.2 NONLINEAR RNNs HAVE LINEAR FEATURE DIRECTIONS IN THE LIMIT OF HIGH SPARSITY

Applying a non-linearity to the hidden state immediately breaks the linear representation hypothesis: each feature would be represented along a (not necessarily smooth) curve rather than a straight line. This massively complicates the study of non-linear recurrence in general.

For the case of linear recurrence, we have shown that for sufficiently sparse input sequences, we can assume that there is at most one non-zero input "in play" within a model's hidden state at any given time. Fundamentally, this was based on the idea that there exists a memory window of length $T$ such that inputs older than $T$ time steps cannot contribute significantly to the current hidden state. This essentially arose from the proof in Appendix D.1 that RNNs with linear recurrence must satisfy the echo state property in order to achieve reasonable loss.

We argue that a similar memory window should be expected in models with nonlinear recurrence. Although it is much harder to prove that the echo state property is a requirement for good performance in non-linear models, there exists plenty of evidence for the reverse statement: echo state networks are, by definition, nonlinear models that satisfy the echo state property (Jaeger, 2002) and have been shown to achieve strong performance on a variety of sequential processing tasks (Aceituno et al., 2020). Clearly, the echo state property is not incompatible with strong task performance and it is not implausible that the linear result – that the echo state property is actually required for tasks like $k$-delay – carries over to the case of nonlinear recurrences, based on these empirical observations. Furthermore, it's well known that RNNs with spectral radius over 1 cause exploding gradients and training instability (Bengio et al., 1994; Hochreiter et al., 2001; Pascanu et al., 2012), and that task dynamics with strong early correlations (producing a spectral radius over 1) result in network instability (Proca et al., 2025). Therefore, we intuitively expect our reasoning to hold for nonlinear recurrence: if the input sequence is made sufficiently sparse, we can approximate the model's behavior by ignoring situations where two or more inputs are non-zero, as historic features are gradually forgotten by the RNN due to the shrinking effect of the spectral radius ($\rho < 1$).

Unrolling the hidden state, we see that an analysis of the general case is intractable:

$$
\begin{aligned}
\boldsymbol{h}_t &= \boldsymbol{w}_x x_t + W_h \operatorname{ReLU}(\boldsymbol{h}_{t-1}) \\
&= \boldsymbol{w}_x x_t + W_h \operatorname{ReLU}(\boldsymbol{w}_x x_{t-1} + W_h \operatorname{ReLU}(\boldsymbol{h}_{t-2})) \\
&\vdots \\
&= \boldsymbol{w}_x x_t + W_h \operatorname{ReLU}(\boldsymbol{w}_x x_{t-1} + W_h \operatorname{ReLU}(\cdots W_h \operatorname{ReLU}(\boldsymbol{w}_x x_2 + W_h \operatorname{ReLU}(\boldsymbol{w}_x x_1)) \cdots))
\end{aligned}
$$

Suppose, however, that for some $s$, only $x_{t-s}$ is non-zero and all other inputs are assumed to be zero, as per our approximation. Recalling that in our setup, $x_{t-s}$ is scalar and non-negative, the hidden state simplifies as follows:

$$
\begin{aligned}
\boldsymbol{h}_t &= \boldsymbol{w}_x(0) + W_h \operatorname{ReLU}(\boldsymbol{w}_x(0) + W_h \operatorname{ReLU}(\cdots W_h \operatorname{ReLU}(\boldsymbol{w}_x x_{t-s} + W_h(0)) \cdots)) && (57) \\
&= W_h \operatorname{ReLU}(W_h \operatorname{ReLU}(\cdots W_h \operatorname{ReLU}(\boldsymbol{w}_x x_{t-s}) \cdots)) && (58) \\
&= W_h \operatorname{ReLU}(W_h \operatorname{ReLU}(\cdots x_{t-s} W_h \operatorname{ReLU}(\boldsymbol{w}_x) \cdots)) && (59) \\
&= x_{t-s} \underbrace{W_h \operatorname{ReLU}(W_h \operatorname{ReLU}(\cdots W_h \operatorname{ReLU}(\boldsymbol{w}_x) \cdots))}_{\boldsymbol{w}_s} && (60) \\
&= \boldsymbol{w}_s x_{t-s} && (61)
\end{aligned}
$$

Though it is not analytically possible to find a simplified expression for the vector $\boldsymbol{w}_s$, it nevertheless is the direction in which the feature $x_{t-s}$ is represented in the limit of sparsity, as $p \to 0$. Hence, in the extremely sparse regime, the linear representation hypothesis holds for this model. This is not a trivial result; it relies on both the piecewise linearity of ReLU for non-negative inputs and our definition of $\boldsymbol{h}_t$ as the hidden state prior to application of ReLU.

## F.3 Simulations across $k$ and sparsity for nonlinear recurrence

In Figure 11 we provide additional simulations of the solutions learned by optimal nonlinear RNNs for different values of $k$ and sparsity $1 - p$. In the nonlinear RNN, we can observe how the model learns to exploit the interference-free space as sparsity increases and to implement sharp forgetting.

Figure 11: **Solutions to $k$-delay learned by nonlinear RNNs.** Rows correspond to different values of $k$ and columns correspond to different sparsity levels. The $\boldsymbol{w}_s$ are plotted after applying a conformal linear transformation such that the $y$-component of each $\boldsymbol{w}_s$ is $\boldsymbol{w}_y^\top \boldsymbol{w}_s$, and $\boldsymbol{w}_x = \boldsymbol{w}_{s=0}$ points towards positive $x$. Thus the interference-free half-space is simply given by $y < 0$. As before, the output feature, $\boldsymbol{w}_{s=k}$, is marked with a star. Note that the plots vary significantly in scale, so it is not meaningful to compare the magnitude of a particular $\boldsymbol{w}_s$ vector between different plots.

# G   Higher dimensional results

## G.1   Decomposing the loss for vector inputs and outputs ($N_x > 1$)

We can repeat the decomposition in Appendix B to obtain an expression for the expected value of the squared-error loss incurred by linear models in the case of vector inputs and outputs ($N_x = N_y > 1$). We recall that $X_t, \hat{Y}_t \in \mathbb{R}^{N_x}$, $W_s \in \mathbb{R}^{N_h \times N_x}$ and $W_y \in \mathbb{R}^{N_h \times N_x}$ and proceed as before:

$$\mathbb{E}[\mathcal{L}] = \mathbb{E}\left[ \sum_{t=1}^{k} \left\| 0 - \hat{Y}_t \right\|^2 + \sum_{t=k+1}^{T} \left\| X_{t-k} - \hat{Y}_t \right\|^2 \right] \tag{62}$$

$$= \sum_{t=1}^{k} \mathbb{E}\left[ \left\| \hat{Y}_t \right\|^2 \right] + \sum_{t=k+1}^{T} \mathbb{E}\left[ \left\| X_{t-k} - \hat{Y}_t \right\|^2 \right] \tag{63}$$

$$= \sum_{t=1}^{k} \mathbb{E}\left[\left\|\hat{Y}_t\right\|^2\right] + \sum_{t=k+1}^{T} \mathbb{E}\left[\|X_{t-k}\|^2 - 2X_{t-k}^\top \hat{Y}_t + \left\|\hat{Y}_t\right\|^2\right] \tag{64}$$

$$= \sum_{t=k+1}^{T} \left(\mathbb{E}\left[\|X_{t-k}\|^2\right] - 2\mathbb{E}\left[X_{t-k}^\top \hat{Y}_t\right]\right) + \sum_{t=1}^{T} \mathbb{E}\left[\left\|\hat{Y}_t\right\|^2\right] \tag{65}$$

$$= \sum_{t=k+1}^{T} \left(\mathbb{E}\left[\|X_{t-k}\|^2\right] - 2\mathbb{E}\left[X_{t-k}^\top \sum_{s=0}^{t-1} W_y^\top W_s X_{t-s}\right]\right) + \sum_{t=1}^{T} \mathbb{E}\left[\left\|\sum_{s=0}^{t-1} W_y^\top W_s X_{t-s}\right\|^2\right] \tag{66}$$

We extend the temporal sparsity assumption to assume that each input feature follows the same distribution, so that $X_t^{(i)} = B_t^{(i)} U_t^{(i)}$ with $B_t^{(i)} \sim$ Bernoulli($p$), $U_t^{(i)}$ is identically distributed according to any distribution and $\left\{B_t^{(i)}\right\} \cup \left\{U_t^{(i)}\right\}$ are mutually independent.

Then, with $\mu := \mathbb{E}\left[U_t^{(i)}\right]$ and $\nu := \mathbb{E}\left[\left(U_t^{(i)}\right)^2\right]$, we can simplify the first expectation to

$$\mathbb{E}\left[\|X_{t-k}\|^2\right] = \sum_{i=1}^{N_x} \mathbb{E}\left[\left(X_{t-k}^{(i)}\right)^2\right] = \sum_{i=1}^{N_x} \mathbb{E}\left[\left(B_{t-k}^{(i)}\right)^2 \left(U_{t-k}^{(i)}\right)^2\right] = \sum_{i=1}^{N_x} p\nu = N_x p\nu \tag{67}$$

In computing the second expectation, we must handle the case of $s = k$ separately:

$$\mathbb{E}\left[X_{t-k}^\top \sum_{s=0}^{t-1} W_y^\top W_s X_{t-s}\right]$$

$$= \mathbb{E}\left[\mathrm{tr}\left(\sum_{s=0}^{t-1} X_{t-k}^\top W_y^\top W_s X_{t-s}\right)\right] \tag{68}$$

$$= \sum_{s=0}^{t-1} \mathrm{tr}\left(\mathbb{E}\left[X_{t-k}^\top W_y^\top W_s X_{t-s}\right]\right) \tag{69}$$

$$= \sum_{s=0}^{t-1} \mathrm{tr}\left(W_y^\top W_s \mathbb{E}\left[X_{t-s} X_{t-k}^\top\right]\right) \tag{70}$$

$$= \mathrm{tr}\left(W_y^\top W_k \mathbb{E}\left[X_{t-k} X_{t-k}^\top\right]\right) + \sum_{s \neq k}^{t-1} \mathrm{tr}\left(W_y^\top W_s \mathbb{E}\left[X_{t-s} X_{t-k}^\top\right]\right) \tag{71}$$

$$= \mathrm{tr}\left(W_y^\top W_k \mathbb{E}\left[X_{t-k} X_{t-k}^\top\right]\right) + p^2 \mu^2 \sum_{s \neq k}^{t-1} \mathrm{tr}\left(W_y^\top W_s \mathbf{1}\mathbf{1}^\top\right) \tag{72}$$

$$= \left[p^2 \mu^2 \mathbf{1}^\top W_y^\top W_k \mathbf{1} + \left(p\nu - p^2 \mu^2\right) \mathrm{tr}\left(W_y^\top W_k\right)\right] + p^2 \mu^2 \sum_{s \neq k}^{t-1} \mathbf{1}^\top W_y^\top W_s \mathbf{1} \tag{73}$$

$$= \left(p\nu - p^2 \mu^2\right) \mathrm{tr}\left(W_y^\top W_k\right) + p^2 \mu^2 \sum_{s=0}^{t-1} \mathbf{1}^\top W_y^\top W_s \mathbf{1} \tag{74}$$

where we make use of the all-ones vector $\mathbf{1} = [1, 1, 1, \dots]^\top \in \mathbb{R}^{N_x}$.

Finally, the third expectation simplifies as follows:

$$\mathbb{E}\left[\left\|\sum_{s=0}^{t-1} W_y^\top W_s X_{t-s}\right\|^2\right]$$

$$= \mathbb{E}\left[\left(\sum_{s=0}^{t-1} W_y^\top W_s X_{t-s}\right)^\top \left(\sum_{s'=0}^{t-1} W_y^\top W_{s'} X_{t-s'}\right)\right] \tag{75}$$

$$= \mathbb{E}\left[\left(\sum_{s=0}^{t-1} X_{t-s}^\top W_s^\top W_y\right)\left(\sum_{s'=0}^{t-1} W_y^\top W_{s'} X_{t-s'}\right)\right] \tag{76}$$

$$= \sum_{s=0}^{t-1}\sum_{s'=0}^{t-1} \mathbb{E}\left[X_{t-s}^\top W_s^\top W_y W_y^\top W_{s'} X_{t-s'}\right] \tag{77}$$

$$= \sum_{s=0}^{t-1}\sum_{s'=0}^{t-1} \mathrm{tr}\left(W_s^\top W_y W_y^\top W_{s'} \mathbb{E}\left[X_{t-s'} X_{t-s}^\top\right]\right) \tag{78}$$

$$= \sum_{s=0}^{t-1} \mathrm{tr}\left(W_s^\top W_y W_y^\top W_s \mathbb{E}\left[X_{t-s} X_{t-s}^\top\right]\right) + \sum_{s\neq s'}^{t-1} \mathrm{tr}\left(W_s^\top W_y W_y^\top W_{s'} \mathbb{E}\left[X_{t-s'} X_{t-s}^\top\right]\right) \tag{79}$$

$$= \left[p^2\mu^2 \sum_{s=0}^{t-1} \mathrm{tr}\left(W_s^\top W_y W_y^\top W_s \mathbf{1}\mathbf{1}^\top\right) + \left(p\nu - p^2\mu^2\right)\sum_{s=0}^{t-1} \mathrm{tr}(W_s^\top W_y W_y^\top W_s)\right] \tag{80}$$

$$+ p^2\mu^2 \sum_{s\neq s'}^{t-1} \mathrm{tr}\left(W_s^\top W_y W_y^\top W_{s'} \mathbf{1}\mathbf{1}^\top\right) \tag{81}$$

$$= \left(p\nu - p^2\mu^2\right)\sum_{s=0}^{t-1} \mathrm{tr}(W_s^\top W_y W_y^\top W_s) + p^2\mu^2 \sum_{s=0}^{t-1}\sum_{s'=0}^{t-1} \mathbf{1}^\top W_s^\top W_y W_y^\top W_{s'} \mathbf{1} \tag{82}$$

$$= \left(p\nu - p^2\mu^2\right)\sum_{s=0}^{t-1} \left\|W_y^\top W_s\right\|_F^2 + p^2\mu^2 \left\|\sum_{s=0}^{t-1} W_y^\top W_s \mathbf{1}\right\|^2 \tag{83}$$

where $\|\cdot\|_F$ is the Frobenius norm.

Putting all the terms together yields

$$\mathbb{E}[\mathcal{L}] = \sum_{t=k+1}^{T}\left(N_x p\nu - 2\left[\left(p\nu - p^2\mu^2\right)\mathrm{tr}(W_y^\top W_k) + p^2\mu^2 \sum_{s=0}^{t-1}\mathbf{1}^\top W_y^\top W_s\mathbf{1}\right]\right) \tag{84}$$

$$+ \sum_{t=1}^{T}\left[\left(p\nu - p^2\mu^2\right)\sum_{s=0}^{t-1}\|W_y^\top W_s\|_F^2 + p^2\mu^2\|\sum_{s=0}^{t-1} W_y^\top W_s\mathbf{1}\|^2\right] \tag{85}$$

## G.2 ANALYZING HIGHER-DIMENSIONAL HIDDEN STATES ($N_h > 2$)

So far, for the purposes of feature geometry, we have restricted the hidden state of our models to 2 dimensions. In this section, we demonstrate that the results we have found generalize well to RNNs with higher-dimensional hidden states. In particular, we find that the interference-free space is not only present, but very well exploited in higher dimensions.

Figure 12 shows the results of training higher-dimensional nonlinear RNNs on the 2-delay task with vector inputs ($N_x = 10$). Even in the 10-dimensional hidden state, there is still significant superposition occurring: a 10-dimensional hidden state can only represent 10 features orthogonally, whereas the task requires 10 features to be held in memory over $k + 1 = 3$ timesteps, equivalent to 30 features being compressed into 10 dimensions.

The 10-dimensional case exhibits very little projection interference – most cells are blue except on the $s = 2$ diagonal, where we expect them to be red. This shows that most activations lie within the interference-free space, where their projection onto every readout vector is negative (and therefore their contribution to projection interference is zero due to the ReLU activation function). We believe this is a strong result that demonstrates that the interference-free space is a significant driver of feature geometry, even in – or perhaps especially in – higher-dimensional hidden states.

Based on these findings, we extend this idea to larger RNNs, developing metrics to quantify the degree to which these networks exhibit the geometry we would expect based on our study. In particular, we measure the mean non-output feature direction projections onto the readout (i.e.,

(a) 2-dimensional hidden state

(b) 5-dimensional hidden state

(c) 10-dimensional hidden state

Figure 12: **Projections of feature directions onto readouts in higher-dimensional hidden states.** Each model was trained on the 2-delay task over 10 features (hence there are also 10 readouts). Each plot shows the values of $W_y^\top W_s$ – the projections of feature directions corresponding to $s$-timestep-old inputs onto each of the readout vectors. Specifically, each cell $(i, j)$ represents the value of $W_y^{(i)\top} W_s^{(j)}$, where $W_y^{(i)}$ is the $i$-th column of $W_y$ and $W_s^{(j)}$ is the $j$-th column of $W_j$. Cells that are red correspond to a positive projection onto the readout – this is desired only on the diagonal of the $s = 2$ panel, where features are being read out into the correct outputs at the correct time; red elsewhere represents projection interference. Blue cells have a negative projection onto the readout and, due to the ReLU, do not contribute to projection interference for their row's readout. **(a)** A 2-dimensional hidden state is only able to represent 2 features well and ignores the rest – a clear example of the 'all-or-none' strategy discussed in Section 4.5. This result is equivalent to the $k = 2$ panel of Figure 5. **(b)** A 5-dimensional hidden state performs much better, representing many more of the features, albeit with some interference. **(c)** A 10-dimensional hidden state performs very well, representing all the features with almost no interference.

Figure 13: **Mean projections of output and non-output feature directions onto readouts colored by loss in higher-dimensional hidden states.** We train 1000 models with a hidden size of 100 on the 2-delay task with 75 features. The x-axis indicates the mean of the non-output feature direction projections onto the readout (i.e., $W_y^\top W_{s \neq k}$). This is the analogous to the mean of the $s \neq 2$ matrices in Figure 12. The y-axis indicates the mean of the output feature direction projections onto the readout (i.e., $W_y^\top W_{s=k}$), which is analogous to the diagonal of the $s = 2$ matrix in Figure 12. By looking at the plot, we see that all models learn to have non-output feature directions that have negative projections onto the readout (negative x-axis values), utilizing the interference-free space. We also see that performance is correlated with the RNN's ability to positively project the output feature direction onto the readout, indicating that the optimal models are using the the interference-free space to minimize projection interference with this output feature.

$\text{mean}(W_y^\top W_{s \neq k})$), which allows us to quantify the degree to which non-output feature directions group within the interference-free space. In particular, this value should be negative if the largest non-output feature directions are grouping within the interference-free space. This is analogous to the mean of the $s \neq 2$ matrices in Figure 12.

We also measure the mean of the output feature direction projections onto the readout (i.e., $\text{mean}(\text{diag}(W_y^\top W_{s=k}))$), which quantifies whether the appropriate features are projected onto the readout at the correct time to perform the task. We would expect this value to be positive for models that successfully do this. This is analogous to the mean of the diagonal of the $s = 2$ matrix in Figure 12.

We train 1000 RNNs with a hidden size of 100 on a 2-delay task with 75 features and plot the results in Figure 13, where each point is representative of a single model and is colored by the final loss it achieves. By looking at the figure, we see that all models learn to group the largest non-output feature directions in the interference-free space (indicated by the negative-valued x-axis). We also see that the best-performing models (lowest loss) learn to successfully project the output feature direction onto the readout (indicated by the positive-valued y-axis), indicating that these (optimal) models are outputting the correct features at the correct timestep and using the interference-free space to minimize projection interference with this output feature. Our results pertaining to optimal model geometry are therefore corroborated by this experiment.

In our experiment, there are models that do not learn the optimal solution (i.e., they have a negative output feature direction projection onto the readout; the yellow/light green points in Figure 13). We suspect that this has to do with the gradient backpropagation through ReLU, as the negative projections can get clipped to zero by the ReLU nonlinearity, preventing a learning signal from going through. Indeed, we do not use any additional methods to assist with training in this experiment.

Moreover, if we train instead with a leaky ReLU activation (permitting gradients to backpropagate to negative projections), there are no models with a negative mean output projection (all models learn the optimal solution metric-wise). We note however that the extension of our work to other activation functions is non-trivial and beyond the scope of this work.

## H    TASKS WITH RANDOM DELAY

In this paper we primarily focus on tasks with a fixed $k$-delay. Here, we instead consider the effect of training on a task with random delay. The task we consider is identical to the $k$-delay task in that the RNN must reproduce the input sequence after $k$ timesteps, but now $k$ is random for each training sample ($k \sim$ Uniform(0,10)). One dimension of the input corresponds to the cue, which remains 0 until the randomly selected $k$, after which it is set to 1 and the RNN is tasked with outputting the sequence, corresponding to the input from the $t - k, \forall t > k$. We keep the input and output dimensions of the network the same– hence, with the addition of a cue input dimension, we also have an extra output dimension that is weighted with 0 importance in the loss. We mask the network output in the loss until after the cue is turned on, such that earlier outputs do not contribute to the loss. We train RNNs of each architecture (linear: Figure 14, SSM: Figure 15, nonlinear: Figure 16) and visualize how the feature geometry changes as the number of input features (e.g., via input dimensions) is varied (the rows) and sparsity is varied (the columns). For each configuration, we train 500 models and plot the one achieving the lowest loss.

Although we are cautious about overinterpreting these plots, we provide a preliminary analysis. The results seem to suggest some intermediate geometry between spatial superposition and temporal superposition. Indeed, the notion of time-dependency here marks a departure from the rest of the paper in that the *sequential ordering* of features is important for the task, but a *time-dependent* output is not (in the sense of a fixed $k$). We see that many RNNs form solutions where feature directions lie on a shrinking line (instead of a spiral sink), with a fixed point at the origin (corresponding to 'forgetting'). 'Age' (for sequential ordering) is still partially encoded by the magnitude of the feature direction on the line. RNNs also appear to be implementing some form of spatial superposition in some cases, partitioning the activation space for several different features; this behavior clearly contrasts from Figure 5, which, for many input features and delays of up to 10, would only choose to represent one feature. However, we also often see a collapse of several feature directions onto the same line. In fact, although we study up to 7 input features, the models typically converge to approximately 2-3 principle directions. We can also see how for 2 features, RNNs learn to either represent these features approximately orthogonally, or place the features in opposite directions.

Remarkably, in the SSM with high sparsity, we recover the pentagon of 5 features characteristic of spatial superposition. We suspect that the SSMs geometric strategies are a result of each feature direction placing itself in the interference-free space of the other feature directions' readouts (hence the spiraling, perpendicular, and pentagon shapes).

We note that these results are preliminary and that more work, both theoretical and empirical, is needed to characterize the effect of random delays on learned feature geometry. A limitation of this particular set of experiments is the lack of expressivity of the models we consider. Although they learn some interesting structure suggesting the use of the interference-free space/a superposition-like strategy, RNNs with 2 neurons do not have the expressivity to perform a cued random-delay task perfectly.

Previous work has shown that RNNs trained on tasks with random delays exhibit persistent activity in the form of fixed point attractors (Orhan & Ma, 2019; Liu et al., 2021; Xie et al., 2022b), which has been related to neural activity. In the results we show here, we do not find a clean example of (non-trivial) fixed point attractors forming. This difference may be due to the fact that the task we study requires the reproduction of a sequence relying on temporal ordering which may be challenging (if not impossible) to implement with a fixed point solution (at least in the architectures we consider here). A critical component of temporal superposition, as we've introduced it here, is the importance of time-dependency in data (inputs and outputs): temporal superposition occurs *because* timing is important and inputs and outputs at different timesteps are treated as unique features (represented uniquely). Tasks based on random delays mark a slight departure, as the precise timing relationship of input-to-output is modified (such that inputs and outputs may not always be 'unique features'). This is also true for other tasks where feature timing is unimportant. However, since the task we

consider here still requires the reproduction of a sequence based on temporal ordering, such ordering still appears to affect representational geometry. Moreover, depending on the tasks an RNN is trained on and its architecture, it may implement a combination of different solutions for different computations/subtasks based on temporal superposition, fixed point solutions, or otherwise (Driscoll et al., 2024). Finally, as stated earlier, we emphasize that our 2D models lack expressivity for more sophisticated solutions and we use a specific version of a random-delay task based on masking the loss prior to cue, followed by a sustained cue after the delay. It's possible that other implementations of a random-delay task may produce other solutions.

Figure 14: **Solutions to random-delay learned by linear RNNs.** Rows correspond to different numbers of input features (1,2,3,5,7) and columns correspond to different sparsity levels (0, 0.7, 0.9, 0.97, 0.99). Each feature is indicated by a separate marker, and 'age' in the network is indicated by color (*purple is new; yellow is old*).

Figure 15: **Solutions to random-delay learned by SSMs.** Rows correspond to different numbers of input features and columns correspond to different sparsity levels. Each feature is indicated by a separate marker, and 'age' in the network is indicated by color (*purple is new; yellow is old*).

Figure 16: **Solutions to random-delay learned by nonlinear RNNs.** Rows correspond to different numbers of input features and columns correspond to different sparsity levels. Each feature is indicated by a separate marker, and 'age' in the network is indicated by color (*purple is new; yellow is old*).

# I  EXPERIMENTAL DETAILS

Code to reproduce all experiments and figures is available at `https://github.com/kashparty/iclr-rnn-superposition`.

## I.1  NUMBER OF MODELS THAT DEVELOP THE FEATURE GEOMETRIES DISCUSSED

To verify that a reasonable number of models actually achieve the kinds of feature geometry discussed in our work, we train 1000 models of each architecture for various delay and sparsity values (Table 1). We use a set of heuristics for each architecture to quantify whether models exhibit characteristics of their expected feature geometries based on our observations. For linear RNNs, we expect the spectral radius to be reasonably large but below 1 and the task-relevant features to rotate through the plane less than $180°$: $0.5 < \rho(W_h) < 1, 5° < k|\arg(\lambda)| < 180°$, where $\lambda$ is an eigenvalue of $W_h \in \mathbb{R}^{2 \times 2}$. For SSMs, we expect the task-relevant feature direction vectors to instead rotate through more than $180°$: $0.5 < \rho(W_h) < 1, k|\arg(\lambda)| > 180°$. For nonlinear RNNs, we quantify whether the output feature direction projects positively onto the readout and other feature directions

project negatively (tolerating some small positive projection): $\boldsymbol{w}_y^\top \boldsymbol{w}_{s=k} > 0, \boldsymbol{w}_y^\top \boldsymbol{w}_s < 0.1$ for $0 < s \leq k$.

We find that a substantial number of models satisfy our heuristics. We note that optimizing with a two-dimensional hidden space is very challenging for gradient descent. Indeed, in the original work on toy models of superposition (Elhage et al., 2022), the authors also study a 2-dimensional hidden state and report that they fit each model multiple times and take the solution with the lowest loss due to these optimization challenges.

In the case of the linear architecture, the number of models that achieve the expected "spiral sink" feature geometry decreases with increasing $k$ – this is simply because the task becomes too challenging for such a simple architecture to learn well; in many cases, models resort to oscillatory behaviour that achieves suboptimal but lower-than-baseline loss. In contrast, over 40% of SSMs trained on larger $k$ and under high sparsity learn the expected "spiral sink" solution. For the non-linear model, using a rudimentary heuristic, we find that a reasonable proportion of models can be clearly said to have learned a feature geometry in which they exploit the interference-free space as discussed in Section 4.4.

|       |         | Number of models (%) | | |
|-------|---------|--------|------|-----------|
| $k$   | $1-p$   | Linear | SSM  | Nonlinear |
| 2     | 0.9     | 26.5   | 31.7 | 31.4      |
| 2     | 0.97    | 30.8   | 20.6 | 26.8      |
| 2     | 0.99    | 27.0   | 28.5 | 27.6      |
| 3     | 0.9     | 21.0   | 41.3 | 18.8      |
| 3     | 0.97    | 18.2   | 32.8 | 15.6      |
| 3     | 0.99    | 19.0   | 33.8 | 13.8      |
| 5     | 0.9     | 19.4   | 53.5 | 8.20      |
| 5     | 0.97    | 18.2   | 48.0 | 10.3      |
| 5     | 0.99    | 16.0   | 45.3 | 9.40      |
| 7     | 0.9     | 14.3   | 47.1 | 12.3      |
| 7     | 0.97    | 11.4   | 42.5 | 15.6      |
| 7     | 0.99    | 11.4   | 41.5 | 13.0      |

Table 1: **Percentage of models that exhibit characteristics of optimal geometry**, as defined by the following heuristics. *Linear:* we expect a reasonable spectral radius and for the feature direction vectors to rotate through less than $180°$ between input and output, so we count the number of models satisfying $0.5 < \rho(W_h) < 1$ and $5° < k|\arg(\lambda)| < 180°$, where $\lambda$ is an eigenvalue of $W_h \in \mathbb{R}^{2\times 2}$. *SSM:* similar to the linear case, but here we expect the feature direction vectors to rotate through more than $180°$ between input and output, so we check for $0.5 < \rho(W_h) < 1$ and $k|\arg(\lambda)| > 180°$. *Nonlinear:* the optimal geometry occurs when all but the output feature direction vector lie in the interference-free space; only one feature direction vector should project positively onto the readout vector, so we check for $\boldsymbol{w}_y^\top \boldsymbol{w}_k > 0$ and $\boldsymbol{w}_y^\top \boldsymbol{w}_s < 0.1$ (tolerating some small positive projections) for $0 < s \leq k$.

## I.2 WEIGHT TYING

For visual clarity, in Figure 1, Figure 3 and Figure 5, we set the readout vector $\boldsymbol{w}_y \coloneqq \boldsymbol{w}_{s=k}$ (or, in the case of vector inputs and outputs, $W_y \coloneqq W_{s=k}$). This means that, in these cases, $\boldsymbol{w}_y$ (or $W_y$) is not a separate trainable parameter of the model and is instead entirely determined by the parameters $W_h$ and $\boldsymbol{w}_x$ (or $W_x$). There are two reasons for doing this: first, it eliminates the need for a separate readout vector to be shown, making the plots neater; second, it encourages features in spatial superposition to arrange into regular polygons (e.g. Figure 1a). In fact, this is merely an extension of the weight tying used by Elhage et al. (2022), where the authors set $W_y \coloneqq W_x^\top$ for the same reasons. Our weight tying is identical to theirs in the $k = 0$ case.

We note that this is just a visualization trick and, to avoid doubt, Figure 17 plots spatial and temporal superposition with the readout weight untied (i.e., with $\boldsymbol{w}_y$ as a separate trainable parameter). Additionally, all figures in the appendix are with weights untied.

(a) Spatial superposition
($W_y$ untied, hidden)

(b) Spatial superposition
($W_y$ untied, shown)

(c) Temporal superposition
($\boldsymbol{w}_y$ untied, shown)

Figure 17: **Spatial and temporal superposition with untied readouts.** Readout vectors are shown as dashed black arrows. **(a)** If $W_y$ is untied from $W_{s=k}$, spatial superposition of 5 features no longer forms a regular pentagon. This is also true of the results in Elhage et al. (2022). **(b)** Untied results are more visually cluttered as both the $W_s$ and the $W_y$ vectors need to be plotted, and these sets of vectors can overlap or be at different scales. **(c)** The difference between tied and untied results is most significant when spatial superposition is involved; for purely temporal superposition, the difference is minimal. Nevertheless, Appendix Figures 10 and 11 each contain many examples of the feature geometry of recurrent models with untied weights.

## I.3 DATA GENERATION

Unless otherwise specified, all experiments used $U_t \sim \text{Uniform}[0, 1]$, so

$$\mu = \mathbb{E}[U_t] = \frac{1}{2}, \qquad \nu = \mathbb{E}[U_t^2] = \frac{1}{3}$$

When generating data to train a model that contains no bias terms, we only include sequences that contain a non-zero input at some timestep. This is an optimization that exploits the fact that bias-free models, by definition, cannot produce non-zero output for inputs that are all-zero; we can thus safely ignore all-zero sequences, as the model will always produce the desired output $\hat{\boldsymbol{y}} = \boldsymbol{x}_{t-k}$ in those cases, the loss will be zero and so no gradient will be backpropagated for such sequences. This makes training much quicker: in cases where sparsity is extremely high (e.g. $1 - p = 0.999$), many generated sequences are all-zero. Combined with the fact that we use a constant learning rate, this optimization has no impact on the training of the model.

The specific shape and sparsity of data generated varies by experiment and is discussed below.

## I.4 MODEL DEFINITION, INITIALIZATION AND TRAINING

As per Equation (1), there are two activation functions $\sigma_h$ and $\sigma_y$ that can be set as follows to achieve a linear RNN, SSM or nonlinear RNN:

- **Linear RNN:** $\sigma_h = \text{id}$, $\sigma_y = \text{id}$
- **SSM:** $\sigma_h = \text{id}$, $\sigma_y = \text{ReLU}$
- **Nonlinear RNN**: $\sigma_h = \text{ReLU}$, $\sigma_y = \text{ReLU}$

In all experiments, the model weights $W_x \in \mathbb{R}^{N_h \times N_x}$ and $W_h \in \mathbb{R}^{N_h \times N_h}$ were initialized using Xavier normal distributions. In cases where the readout weights $W_y \in \mathbb{R}^{N_h \times N_y}$ are *not* tied to $W_h^k W_x$, we initialize $W_y$ using a Xavier normal distribution.

In practice, we often trained many models in parallel to make efficient use of GPU compute. Data was batched into 1000 batches; for each batch, we computed each model's average training loss over an entire sequence (weighting the contribution of each feature to the loss by the feature's importance) and backpropagated from this value. We maintained an exponential moving average (EMA) of this

value for each model according to the update equation:

$$\ell_{\text{EMA}} \leftarrow \ell_{\text{EMA}} + 0.01(\ell - \ell_{\text{EMA}})$$

After training on all 1000 batches, the final value of $\ell_{\text{EMA}}$ was used to compare models; unless otherwise stated, the model that achieved the lowest EMA training loss was selected for plotting.

In all experiments, we used the `AdamW` optimizer with a constant learning rate of $\text{lr} = 5 \times 10^{-3}$.

Feature importance is attributed by weighting the loss according to the importance value per feature, as in Elhage et al. (2022). In other words $\mathcal{L} = \sum_{t=1}^{T} I(\boldsymbol{x}_{t-k}) \|\boldsymbol{x}_{t-k} - \hat{\boldsymbol{y}}_t\|^2$, where $I(\boldsymbol{x}_{t-k})$ is a scalar sum of the corresponding importance values of $\boldsymbol{x}_{t-k}$.

### I.5 FIGURE DETAILS

#### I.5.1 FIGURE 1

**Panel (a):** 100 nonlinear RNNs with 2-dimensional hidden states were trained on 10k non-zero sequences of length 10 timesteps each. The task was 0-delay, so $y_t = x_t$, making this equivalent to the first task used in Elhage et al. (2022). The input sequences had sparsity 0.99 (so $p = 0.01$) and contained 5 features, $\{A, B, C, D, E\}$ with importances $\{1, 0.97, 0.97^2, 0.97^3, 0.97^4\}$ respectively.

**Panel (b):** Identical to panel (a), except trained on the $k = 5$ task instead of $k = 0$. To decrease visual clutter, only the most important feature (A) was plotted. Note that due to the random nature of data generation and training, it is not guaranteed that feature A will *always* be prioritized over the other, less important features (e.g. B might instead be prioritized), but a single feature is almost always prioritized over all others and A is the most common choice.

#### I.5.2 FIGURES 2 AND 6

A linear RNN with 2-dimensional hidden state was trained on 50k non-zero scalar input sequences, each of length 20 time steps and sparsity 0.9. The task was 3-delay. At each step of training, we computed the values of $\boldsymbol{w}_y^\top \boldsymbol{w}_s$ for $0 \leq s < 12$ and used these to calculate the contribution of each of the four terms in equation 5 to the loss, as plotted in Figure 2. In Figure 6, we additionally plot the eigenvalues of $W_h$ and the corresponding $\boldsymbol{w}_s$ geometry at different points in training, as well as the singular values.

#### I.5.3 FIGURE 3

For each architecture (linear, SSM, nonlinear), 100 models with 2-dimensional hidden states were trained on the 5-delay task using 10k non-zero sequences of length 25 timesteps each. We used a sparsity level of 0.99, so $p = 0.01$. We plotted illustrative examples of models achieving the lowest loss.

#### I.5.4 FIGURE 4

Here we trained models on the $k = 7$ task. We swept through 200 uniformly spaced sparsity values in the interval $[0.5, 1)$; at each sparsity level, we trained 1000 SSMs with 2-dimensional hidden states on 10k non-zero scalar input sequences of length 25 timesteps each. For each sparsity level, we took the best 50 models (top 5%) in terms of lowest EMA training loss. The mean value and standard deviation of $\rho$ and $k\theta$ across these 50 best models for each sparsity level was plotted, thus indicating the "optimal" $\rho$ and $k\theta$ at each value of sparsity.

#### I.5.5 FIGURE 5

This experiment is identical to that for Figure 1, except that $k = 1$, $k = 2$ and $k = 3$ were also included. The heatmap was computed by taking a $2000 \times 2000$ grid of points within the axes and, at each point, computing the sum of its non-negative projections onto the readout vectors (columns of $W_y$). The heatmap therefore visualizes the region in activation space within which projection interference is zero – the interference-free space.

### I.5.6  FIGURE 8 AND FIGURE 9

We swept through 300 uniformly spaced values for $\tau \in [-2, 2]$ and 300 uniformly spaced values for $\delta \in [-2, 2]$. For each pair $(\tau, \delta)$, we parameterized a linear RNN with 2-dimensional hidden state as described in Appendix D.2. Each model was trained on 10k non-zero scalar input sequences of length 20 timesteps each. Results are plotted for each combination of delay and sparsity level $(k, 1-p) \in \{2, 4\} \times \{0.7, 0.9, 0.999\}$.

### I.5.7  FIGURE 10

For each combination of delay and sparsity level, $(k, 1-p) \in \{2, 3, 5, 7\} \times \{0, 0.7, 0.9, 0.97, 0.99\}$, 100 models with 2-dimensional $W_h$, linear recurrence and ReLU readout were trained on 10k input sequences, each of length 25 timesteps. Each plot shows the vectors of the best-performing model (as measured by lowest EMA training loss). The values $\rho$ and $k\theta$ shown under the plots are calculated from the final $W_h$ given its eigenvalues $\lambda_1, \lambda_2 \in \mathbb{C}$ as follows:

$$\rho := \max(|\lambda_1|, |\lambda_2|)$$
$$k\theta := k \arg(\lambda_1)$$

### I.5.8  FIGURE 11

The process was the same as for Figure 10, but with a nonlinear RNN trained instead of an SSM. Due to the nonlinear recurrence, $\rho$ and $k\theta$ were not meaningful values to compute and so were omitted from the figure.

### I.5.9  FIGURE 12

We train 100 nonlinear RNNs with hidden sizes of $N_h = 2, 5, 10$ on the 2-delay task with vector inputs ($N_x = 10$). For the best performing model of each hidden size, we plot the values of $W_y^\top W_s$ for $s \in \{1, 2, 3, 4, 5\}$.

### I.5.10  FIGURE 13

We train 1000 nonlinear RNNs with a hidden size $N_h = 100$ on a 2-delay task with 75 input features. In the plot, for each model, we compute the mean of the output projection onto the readouts (mean(diag($W_y^\top W_{s=k}$))) and the mean of the non-output projection onto the readouts (mean($W_y^\top W_{s\neq k}$)). We additionally color each model by the training loss achieved (purple being lowest, yellow being highest).

### I.5.11  FIGURES 14 TO 16

For each combination of number of features and sparsity level, $(N_x - 1, 1 - p) \in \{1, 2, 3, 5, 7\} \times \{0, 0.7, 0.9, 0.97, 0.99\}$, 500 models of each architecture (linear, SSM, nonlinear) were trained on 10k input sequences, each of length 20 timesteps. Each model is tasked with reproducing the input sequence after $k$ timesteps, but now $k$ is random for each training sample ($k \sim \text{Uniform}(0, 10)$). One dimension of the input corresponds to the cue, which remains 0 until the randomly selected $k$, after which it is set to 1 and the RNN is tasked with outputting the sequence, corresponding to the input from $t - k, \forall t > k$. We keep the input and output dimensions of the network the same, such that we include an extra output dimension that is weighted with 0 importance in the loss. We mask the network output in the loss until after the cue is turned on, such that earlier outputs do not contribute to the loss.

Each plot shows the trajectory of feature directions for each respective input feature through time (purple being new, yellow being old), indicated by a unique marker shape. Each plot visualizes the best-performing model.

