# OpenReview forum: "Temporal superposition and feature geometry of RNNs under memory demands"
_ICLR.cc/2026/Conference — ICLR 2026 Oral_

### Official Review · Reviewer_pihS · 2025-10-16

**Soundness:** 4
**Presentation:** 4
**Contribution:** 3
**Rating:** 8
**Confidence:** 4

**Summary:**

In this paper, the authors introduce a novel concept of temporal superposition in RNNs, the idea that beyond spatial superposition of features, memory demand also poses a pressure for the network to pack features presented at different time steps into limited hidden dimensions.

The authors analyze RNNs on a k-delay task and identify two interference modes: projection interference and composition interference. They analytically decomposed the loss into four interpretable terms,  including the task benefits and the two interference modes, which can be used to explain the learned geometry of feature directions in the hidden state. The authors also identified a phase transition in RNN's representational geometry as a function of the sparsity of features. In their analysis of nonlinear RNNs, the authors provide a geometric perspective on how nonlinear RNNs can implement sharp forgetting while linear RNNs can only implement gradual forgetting.

Finally, the authors also study the interaction between spatial and temporal superposition as a function of memory demand, and discovered that as memory demand increases, the network goes from representing all features equally to representing only the most important feature for all time steps.

**Strengths:**

- This paper presents very interesting results regarding temporal superposition of features in RNNs under memory demand, which I believe can be of broad relevance for the mechanistic interpretability and computational neuroscience community.
- The concept of temporal superposition is novel, and the authors use well-designed experiments to illustrate this effect on three types of RNNs while contrasting the role of nonlinearity in feature representation.
- The authors provide clear, interpretable theory on RNN geometric strategy by decomposing the loss into four terms that correspond to the task performance, correction effect, and two types of interference.
- The paper is exceptionally clear: key concepts are precisely defined, notation is consistent, and figures concretely illustrate the geometric claims, making it easy to follow.

**Weaknesses:**

- The core analysis revolves around the simple k-delay task. While this isolates temporal superposition effectively, it may not capture real-world memory demands involving manipulation, variable delays, or context-dependent processing.
- Most results are in low-dimensional hidden states (Nh=2), with higher dimensions (up to Nh=10) only briefly explored in the appendix. Scaling to realistic model sizes could reveal different behaviors but those were not considered or discussed.

**Questions:**

- The k-delay task effectively isolates memory demands, but as noted in Limitations, it focuses on reproduction rather than manipulation. What preliminary insights do you have on how temporal superposition manifests in tasks with variable delays or manipulation of information?

---

> ### Author Response · Authors · 2025-11-27
> **Author Response to Reviewer pihS**
>
> Thank you for your thoughtful feedback and for reviewing our paper. We are very grateful for your encouraging words and appreciation of our work.
>
> **Weaknesses**
>
> > The core analysis revolves around the simple k-delay task. While this isolates temporal superposition effectively, it may not capture real-world memory demands involving manipulation, variable delays, or context-dependent processing.
>
> We thank the reviewer for raising this important point and agree that it is important to consider more complex task settings. While the k-delay task we consider may be simple, it does allow us to isolate memory capacity and study several aspects of memory: (1) the length of time a memory is stored, (2) the number of (total) features that are stored, and (3) the amount of information contained in each sequence item (i.e., number of data features), which all interact with network capacity. Furthermore, while the storage and reproduction of time-dependent inputs/outputs may be simple, it constitutes the basis of many real-world tasks, suggesting that its understanding may have important implications. We see this as a building block towards understanding how different computations interact, are composed, and are represented (Driscoll et al., 2024).
>
> Nevertheless, complex computations such as manipulation of information, variable delays, and context-dependent processing are certainly all important aspects of RNNs. While our work does not analyze all of these settings, we believe that by studying how time dependency/memory demand and capacity interact to shape feature geometry, we can build an overall better understanding of how RNNs operate. We believe that the generalizable insight from our work is that superposition (non-orthogonality of feature representations) occurs in capacity-constrained RNNs and is uniquely affected by both “space” and timing/memory demands, which interact and present representational tradeoffs, some of which we characterize here. Furthermore, we show that temporal superposition is critically a result of representing time– indicating that time-dependent input and output drive this behavior. It is our hope that these ideas can be built on to better understand the mechanisms and representational strategies employed by RNNs in diverse task settings.
>
> References:
>
> Driscoll et al. Flexible multitask computation in recurrent networks utilizes shared dynamical motifs. *Nature Neuroscience*, 2024.
>
> > Most results are in low-dimensional hidden states (Nh=2), with higher dimensions (up to Nh=10) only briefly explored in the appendix. Scaling to realistic model sizes could reveal different behaviors but those were not considered or discussed.
>
> We thank the reviewer for bringing this up. In an effort to make our work more general, we have now included a new experiment on RNNs of hidden size 100. In particular, we trained 1000 RNNs on a 2-delay task with 75 features and quantified the mean non-output feature direction projection onto the readout (if negative, is indicative that the largest feature directions are grouped in the interference-free space) and the mean output feature direction projection onto the readout (if positive, is indicative that the output feature direction projects onto the readout at the appropriate time). We found that all models had negative non-output feature direction projections onto the readout (using the interference-free space), and that the best performing models had a positive mean output feature direction projection onto the readout (Appendix G.2, Figure 12). We have also generalized our derivation of the expectation of the loss for linear RNNs to include vector inputs and outputs (Appendix G.1).

---

> > ### Author Response · Authors · 2025-11-27
> > **Author Response Continued**
> >
> > > The k-delay task effectively isolates memory demands, but as noted in Limitations, it focuses on reproduction rather than manipulation. What preliminary insights do you have on how temporal superposition manifests in tasks with variable delays or manipulation of information?
> >
> > We thank the reviewer for raising this important and nuanced issue. Indeed, before our experimentation (see below), we expected that in tasks with variable delays that feature directions would no longer encode “age”/time through rotations, but instead stabilize in one direction (as a fixed point) that can be read-out at some variable time, such that $w_{s} \approx w_{s’}$ for $s \neq s’$. This would recover something similar to pure spatial superposition where, for example features A-E are represented simultaneously in separate directions but there is no time-related component. We have now run new experiments with random-delay, and we see something somewhat similar to this prediction. While feature directions tend to stabilize along a line, age is still partially encoded by the scale of the feature direction, which shrinks towards the origin over time. We suspect that the models we study do not develop persistent/stable fixed point solutions in this setting due to the sequential ordering of inputs which is still time-dependent, and perhaps because of their limited expressivity.
> >
> > This highlights how temporal superposition is a result of time-dependence: RNNs represent time (both input timing and output timing) through feature directions. When timing is less important (e.g., with random delay or without ordered stimuli/sequence), these representations of time may be less pronounced or disappear, recovering spatial superposition. In this sense, removing time-dependence from the task encourages the RNN to act similarly to a feedforward network (through fixed point solutions).
> >
> > To test this, we have now included new experiments across all architectures where RNNs are tasked with performing a sequence reproduction with random cued delay in Appendix H with figures. We copy the text here for convenience:
> >
> > “In this paper we primarily focus on tasks with a fixed $k$-delay. Here, we instead consider the effect of training on a task with random delay. The task we consider is identical to the $k$-delay task in that the RNN must reproduce the input sequence after $k$ timesteps, but now $k$ is random for each training sample ($k\sim\text{Uniform(0,10)}$. One dimension of the input corresponds to the cue, which remains 0 until the randomly selected $k$, after which it is set to 1 and the RNN is tasked with outputting the sequence, corresponding to the input from the $t-k, \forall t>k$. We train RNNs of each architecture (linear: Figure 13, SSM: Figure 14, nonlinear: Figure 15) and visualize how the feature geometry changes as the number of input features is varied (the rows) and sparsity is varied (the columns).
> >
> > Although we are cautious about overinterpreting these plots, we provide a preliminary analysis. The results seem to suggest some intermediate geometry between spatial superposition and temporal superposition. Indeed, the notion of time-dependency here marks a departure from the rest of the paper in that the *sequential ordering* of features is important for the task, but a *time-dependent* output is not. We see that most RNNs form solutions where feature directions lie on a shrinking line (instead of a spiral sink), with a fixed point at the origin (corresponding to 'forgetting'). 'Age' (for sequential ordering) is still partially encoded by the magnitude of the feature direction on the line. RNNs also appear to be implementing some form of spatial superposition in some cases, partitioning the activation space for several different features; this behavior clearly contrasts from Figure 5, which, for many input features and delays of up to 10, would only choose to represent one feature. However, we also often see a collapse of several feature directions onto the same line. In fact, although we study up to 7 input features, the models typically converge to approximately 2-3 principle directions. We can also see how for 2 features, most RNNs learn to represent these features approximately orthogonally.
> >
> > Remarkably, in the SSM with high sparsity, we exactly recover the pentagon of 5 features characteristic of spatial superposition. We suspect that the SSMs geometric strategies are a result of each feature direction placing itself in the interference-free space of the other feature directions' readouts (hence the spiraling, perpendicular, and pentagon shapes).”

---

### Official Review · Reviewer_iPfa · 2025-10-30

**Soundness:** 4
**Presentation:** 4
**Contribution:** 3
**Rating:** 8
**Confidence:** 3

**Summary:**

This paper investigates the representation geometry of RNNs, focusing on how time affects the so-called superposition hypothesis.

The authors introduce the concept of temporal superposition, analyze the resulting interferences and their interaction with spatial superposition, and derive analytical results on a simple task.

**Strengths:**

Very well written, organized and easy to follow. The figures helps.

The initial analysis of a simpler model help intuition.

Great related works, very comprehensive.

Important generalization of an hypothesis that has garnered a lot of attention in the community. Significant.

**Weaknesses:**

Technically, the contribution could feel somewhat incremental compared to prior work on spatial superposition. However, the significance is still there.

Many key results are presented only in the Appendix, which reduces the overall impact of the paper. This makes me wonder whether the work might be better suited for a journal that allows a longer format. For instance, Figure 9 is particularly helpful for understanding the core concept but is unfortunately buried in the Appendix.

The analysis with multiple dimensions should also be moved to the main text and discussed in more depth.

**Questions:**

The qualitative results shown in Figures 2 and 3 are a bit unclear: are these plots averaged over 100 runs? or are they representative examples? Could you include error bars to assess variability.

Does the analysis of spatial or temporal superposition provide any insight into how we should train or test our networks? Beyond being conceptually interesting, are there practical consequences of this phenomenon of temporal superposition?

Have there been any empirical observations of temporal superposition in animal data?

Minor: Figure 3 is not cited in the main text.

I’m a bit unclear on why feature A competes with itself through time, since, in effect, the latent space at time t is different from the latent space at time t+1. The network doesn’t hold At and At+1 simultaneously, only ht worries about At and ht+1 worries about At+1. I’m viewing the RNN as unrolled.

Put another way, could temporal superposition also occur in a feedforward neural network, with the layer index playing the role of time? If not, why not?

---

> ### Author Response · Authors · 2025-11-27
> **Author Response to Reviewer iPfa**
>
> Thank you for your thoughtful comments and for reviewing our paper. We appreciate your encouraging words on the impact of our work.
>
> **Weaknesses**
>
> > Many key results are presented only in the Appendix, which reduces the overall impact of the paper. This makes me wonder whether the work might be better suited for a journal that allows a longer format. For instance, Figure 9 is particularly helpful for understanding the core concept but is unfortunately buried in the Appendix.
>
> Thank you for taking the time to go through the results in the Appendix. Indeed, having studied several architectures and varying experiments along several dimensions (sparsity, memory demand), there are many results and visualizations to decide between including in the main text. We’ve made an effort to condense the main results in the figures of the main paper, but it certainly helps to see how these different factors affect the geometry as you state. We’ve now included an additional inset of two panels from Figure 9 into the phase transition figure in an effort to convey how varying sparsity affects feature geometry.
>
> > The analysis with multiple dimensions should also be moved to the main text and discussed in more depth.
>
> We thank the reviewer for their interest in this result. We’ve now added the following paragraph on the analysis to multiple dimensions in the main text:
>
> **Higher-dimensional hidden states.** Up to now, we have restricted the hidden state ($N_h$) of our models to 2 dimensions for easier visualization and interpretability. To extend our setting to higher-dimensional hidden states, we train nonlinear RNNs on 10-dimensional input ($N_x=10$) on the 2-delay task, varying hidden size ($N_h=2,5,10$), and measure the projection of each feature direction onto the readout ($W_y^\top W_s$). Based on our previous results, we would expect $W_y^\top W_{s=2}$ to have a diagonal of positive outputs (corresponding to the output feature directions positively projecting onto the readout: for the correct output at the correct time). Moreover, we would expect the rest of the entries in the matrix (as well as all of $W_y^\top W_{s \neq 2}$) to be negative or 0, lying in the interference-free space. Across all hidden sizes, we see this exact strategy (Figure 11 and Appendix G), with RNNs with larger hidden sizes simply capturing more features along the diagonal of $W_y^\top W_{s=2}$ (i.e., the same all-or-none effect described above). Finally, we quantify this behavior by computing the mean of the non-output feature direction projections onto the readout (i.e., $\text{mean}(W_y^\top W_{s \neq k})$) which should be negative in an optimal model, and the mean of the output feature direction projections onto the readout (i.e., $\text{mean}(\text{diag}(W_y^\top W_{s = k}))$), with should be positive. We train RNNs with hidden size 100 on a 2-delay task with 75 features and find that the best performing models group the largest feature directions into an interference-free space and project the output feature onto the readout at the appropriate time, as predicted (Figure 12).
>
> **Questions**
>
> > The qualitative results shown in Figures 2 and 3 are a bit unclear: are these plots averaged over 100 runs? or are they representative examples? Could you include error bars to assess variability.
>
> We apologize for the lack of clarity. In Figure 2, the plots are representative examples. In Figure 3, the plots were formerly the mean of the 5 best-performant models (lowest loss) out of 1000 trained models. We selected the best 5 models to show what the “optimal” model configuration is in the setting we study. We’ve now updated this figure to instead include the mean and error over the best 50 models, which has higher variance.

---

> > ### Author Response · Authors · 2025-11-27
> > **Author Response Continued**
> >
> > > Does the analysis of spatial or temporal superposition provide any insight into how we should train or test our networks? Beyond being conceptually interesting, are there practical consequences of this phenomenon of temporal superposition?
> >
> > That’s a great question. Our analysis was mainly driven by a motivation to understand how superposition is affected by a new architecture with different constraints that may affect geometry and to use this to better understand the representational strategies learned by RNNs. Nevertheless, it can provide some practical insights into RNN behavior. For example, we can make predictions about how spatial and temporal sparsity of the data will affect the strategy an RNN learns, which in turn can affect how it generalizes to changes in the distribution of the data. Moreover, we can better understand the relationship between different aspects of the task trained on– sparsity, memory demand, and the number of features– and how it relates to capacity, to better predict the size of the network needed to successfully perform the task. It could also potentially help us understand when and why RNNs make certain errors through the concepts of composition and projection interference we introduce here– for example, confusing one feature for another or outputting a feature at the wrong time. Furthermore, work in LLMs has demonstrated how an understanding of superposition can inform approaches to removing features from superposition to provide interpretability of complex models, trace circuits in models, and change their behavior (Templeton et al., 2024, Gao et al. 2024; Lindsey et al., 2025). It’s possible that similar approaches could also be applied to RNNs as we improve our understanding of how superposition behaves in new architectures.
> >
> > References:
> >
> > Templeton, et al. Scaling Monosemanticity: Extracting Interpretable Features from Claude 3 Sonnet. *Transformer Circuits Thread*, 2024.
> >
> > Gao et al. Scaling and evaluating sparse autoencoders. *ICLR*, 2025
> >
> > Lindsey et al. On the Biology of a Large Language Model. *Transformers Circuits Thread*, 2025.
> >
> > > Have there been any empirical observations of temporal superposition in animal data?
> >
> > The setting we consider here departs from the typical neuroscience experimental setting (our task isn’t partitioned into trials; we assume temporal and spatial sparsity and limited capacity/underparameterization), so we are cautious to make any direct parallels or comparisons to animal data. Nevertheless, there are some similarities between the behavior we see and prior analyses of neural data. For example, previous work (Xie et al., 2022) in monkeys trained on sequence working memory tasks has shown that sequence items are represented in distinct neural subspaces, akin to our ‘feature direction’ for input features at each timestep. However, these subspaces are near-orthogonal, which is different from our definition of superposition as non-orthogonal feature directions stemming from limited capacity and sparsity. Our concept of the interference-free space is also similar, for example, to output-null subspaces that allow for preparatory activity (without movement) in the motor cortex (Kaufman et al., 2014).
> >
> > References:
> >
> > Xie et al. Geometry of sequence working memory in macaque prefrontal cortex. *Science*, 2022.
> >
> > Kaufman et al. Cortical activity in the null space: permitting preparation without movement. *Nature Neuroscience*, 2014.
> >
> > > Minor: Figure 3 is not cited in the main text.
> >
> > Thank you for pointing this out. We’ve updated the main text to cite Figure 3.
> >
> > > I’m a bit unclear on why feature A competes with itself through time, since, in effect, the latent space at time t is different from the latent space at time t+1. The network doesn’t hold At and At+1 simultaneously, only ht worries about At and ht+1 worries about At+1. I’m viewing the RNN as unrolled.
> >
> > With each timestep in the recurrence (i.e., from $h_t$ to $h_{t+1}$), the feature $A_t$ input at timestep $t$ stays in the hidden state– it is just transformed by a recurrence. So if there’s an input at timestep $t-1$ ($A_{t-1}$) and an input at timestep $t$ ($A_t$), both of these feature representations will remain in the hidden state (until forgotten), but will be successively transformed with each timestep. In this sense, $A_t$ is not competing with itself, per se, but with feature $A_{t-1}$ that was input at a different timestep.

---

> > > ### Author Response · Authors · 2025-11-27
> > > **Author Response Continued (2)**
> > >
> > > > Put another way, could temporal superposition also occur in a feedforward neural network, with the layer index playing the role of time? If not, why not?
> > >
> > > We suspect that a similar phenomenon could take place in feedforward networks, for example, if certain features are linearly represented in the input but are only needed for computation at the $k$th layer (have some ordering of utility by layer depth), or perhaps if certain features are discarded at a certain depth in the network. A key difference in the RNN setting, however, is the time-dependent input and output, which isn’t applicable to a standard feedforward network without additional inputs to/outputs from each layer.

---

### Official Review · Reviewer_DPAE · 2025-10-31

**Soundness:** 3
**Presentation:** 3
**Contribution:** 2
**Rating:** 6
**Confidence:** 4

**Summary:**

Authors consider a set of linear and nonlinear RNNs trained to perform a K-back task and study the properties of learned solutions, primarily for RNNs with 2 units. For the linear case, authors are able to write down the loss function in terms of four distinct components and name the loss contributions in terms of what they call projection and composition interference. Essentially, this paper has the flavor of applying interpretability methods used in LLM world to the studies of computation performed by RNNs.

**Strengths:**

- The analytical derivation of the loss components is novel, and insightful for training and reverse-engineering RNNs to perform the K-back task.
- The paper is packed with a lot of fun to read results/observations, e.g., the phase transition, the generalization to multi-input case and how all but one inputs get dropped out due to constrained representational capacity.
- To me, this paper was instrumental in realizing how far ahead we need to go to in our LLM interpretability studies compared to what the field has achieved for RNNs. So, it is also a nice toy model and has pedagogical implications.

**Weaknesses:**

- As authors admit themselves, the work focuses primarily on a single task and a very small RNN. There is not necessarily a generalizable insight/evidence that I was able to take away, and that does diminish the contribution.
- I find that the authors are less direct with how far RNN studies have come (specifically, the low-rank RNN studies). In many cases, we can now study the low-dimensional subspaces learned by RNNs and draw the exact flow maps they learn (not just the feature dimensions). In that sense, more modern discussion of low-rank RNNs is desirable.
- It is unclear how many RNNs have developed these representations (especially for nonlinear ones) and how many failed to learn or learned completely different solutions. More rigor in reporting is desirable.

**Questions:**

I only have few questions/comments:

- Eq. (1) is actually the more common RNN architecture in computational neuroscience. Hence, Appendix F.1. is making incorrect claims. Also, the connection of Eq. (1) to other form of vanilla RNN is well known. See [1].

- Please emphasize in the abstract that your main contributions hold exactly for linear RNNs, and then you show that some of the insights do generalize to nonlinear counterparts. However, generalization beyond the particular task is not shown, please state this explicitly as well.

- What makes feature A different than others in Figure 4?

- Figure 5 is very interesting, why is it in Appendix? I would argue it may be the most interesting figure in this paper.

- Are you aware of [2]? If so, how does your work, especially Fig. 5, compare to their findings? In my reading, it seems you also find that geometric restructuring (as Haputhanthri et al. 2024 has defined it) can happen even without emergence of attractors, as your case simply uses a spiral. That seems like a noteworthy connection/extension of prior work as well.

My overall assessment is as follows: This work is limited in ambition and has some issues with rigor, both of which limit its contributions. However, it does introduce an interesting toy model and compared to how far we have come with RNN analyses, it does show how rudimentary the superposition/interpretability ideas in LLMs are. We need to do a lot better. With correct placement into the RNN literature, in which we can now study high-dimensional RNNs and their learned algorithms exactly (see recent low-rank RNN studies), I think this manuscript can be a fun read for the ICLR audience.

[1] https://pubmed.ncbi.nlm.nih.gov/22023194/
[2] https://openreview.net/forum?id=njmXdqzHJq

---

> ### Comment · Reviewer_DPAE · 2025-11-24
>
> While I appreciate that all reviewers provided favorable reviews, which may have led to the decision by authors to not provide a rebuttal, I have raised several concerns in my review that are now left unanswered. At the beginning, I felt confident in authors addressing these with a rebuttal. My original score reflected this confidence, which now is proven to be incorrect. Therefore, I have adjusted my score and confidence to reflect my view of the paper as is.

---

> > ### Author Response · Authors · 2025-11-25
> >
> > We would like to thank the reviewer for their time and effort spent providing a thoughtful review of our work. To be clear, it is absolutely our intention to provide a detailed rebuttal in response to all of the reviewers' concerns, regardless of the ratings achieved. We have been working in the background to conduct further experiments to ensure our responses are properly evidenced and researched - some of the reviewers are interested in non-trivial extensions to our work, which naturally take time to investigate - and we would like to apologize for any misunderstanding that has arisen due to the delay. We thank all the reviewers for their patience in this matter and hope that reviewer DPAE can see that we are taking their concerns very seriously and would reconsider their rating in light of the rebuttal that we shall post very soon.

---

> > > ### Comment · Reviewer_DPAE · 2025-11-25
> > >
> > > Of course, please do! I would be very happy to have my concerns addressed!

---

> ### Author Response · Authors · 2025-11-27
> **Author Response to Reviewer DPAE**
>
> Thank you for your helpful comments and detailed review. We are happy to hear that you found our paper fun-to-read.
>
> **Weaknesses**
> > As authors admit themselves, the work focuses primarily on a single task and a very small RNN. There is not necessarily a generalizable insight/evidence that I was able to take away, and that does diminish the contribution.
>
> We thank the reviewer for raising this important issue. While it’s true that our work focuses on a single task, the $k$-delay has a few advantages. (1) The k-delay task provides a direct way to control memory demand (through the hyperparameter $k$), accounting for time-dependent inputs and outputs. Indeed, it has been used in seminal papers to study memory capacity in RNNs (Jaeger, 2002). (2) The $k$-delay task is easier to work with analytically. Furthermore, the $k$-delay task can be performed by both linear and nonlinear networks (unlike cued random delay which requires nonlinearity), allowing us to study and compare between solutions found in different architectures. (3) The $k$-delay task is also the direct extension of the task considered in Elhage et al., 2022 to the temporal domain. Hence, as we show in Section 4.5, we can even interpolate between spatial and temporal superposition by changing $k$. Being the first paper to introduce the concept of temporal superposition, we wanted to focus on a task that would allow us to compare and contrast with spatial superposition introduced in Elhage et al., 2022.
>
> Moreover, the task treats inputs and outputs as time-dependent, which is what we show produces temporal superposition. We do indeed consider a very small RNN for ease of visualization and interpretability, but we show in the appendix, for example, that RNNs group their features into the interference-free space and time the output projection appropriately in higher-dimensional hidden states. However, we agree with the reviewer that our work is indeed a toy setting. Therefore, in an effort to make our work more general, we have included a new experiment on RNNs of hidden size 100. In particular, we trained 1000 RNNs on a 2-delay task with 75 features and quantified the mean non-output feature direction projection onto the readout (if negative, is indicative that the largest feature directions are grouped in the interference-free space) and the mean output feature direction projection onto the readout (if positive, is indicative that the output feature direction projects onto the readout at the appropriate time). We found that all models had negative non-output feature direction projections onto the readout (using the interference-free space), and that the best performing models had a positive mean output feature direction projection onto the readout (Appendix G.2, Figure 12). We have also generalized our derivation of the expectation of the loss for the linear model to include vector inputs and outputs (Appendix G.1).
>
> References:
>
> Jaeger. Short term memory in echo state networks. *German National Research Institute for Computer Science*, 2002.
>
> Elhage et al. Toy models of superposition. *Transformers Circuit Thread*, 2022.

---

> > ### Author Response · Authors · 2025-11-27
> > **Author Response Continued**
> >
> > Finally, we have now included new experiments across all architectures where RNNs are tasked with performing a sequence reproduction with random cued delay in Appendix H with figures. We copy the text here for convenience:
> >
> > “In this paper we primarily focus on tasks with a fixed $k$-delay. Here, we instead consider the effect of training on a task with random delay. The task we consider is identical to the $k$-delay task in that the RNN must reproduce the input sequence after $k$ timesteps, but now $k$ is random for each training sample ($k\sim\text{Uniform}(0,10)$. One dimension of the input corresponds to the cue, which remains 0 until the randomly selected $k$, after which it is set to 1 and the RNN is tasked with outputting the sequence, corresponding to the input from the $t-k, \forall t>k$. We train RNNs of each architecture (linear: Figure 13, SSM: Figure 14, nonlinear: Figure 15) and visualize how the feature geometry changes as the number of input features is varied (the rows) and sparsity is varied (the columns).
> >
> > Although we are cautious about overinterpreting these plots, we provide a preliminary analysis. The results seem to suggest some intermediate geometry between spatial superposition and temporal superposition. Indeed, the notion of time-dependency here marks a departure from the rest of the paper in that the *sequential ordering* of features is important for the task, but a *time-dependent* output is not. We see that most RNNs form solutions where feature directions lie on a shrinking line (instead of a spiral sink), with a fixed point at the origin (corresponding to 'forgetting'). 'Age' (for sequential ordering) is still partially encoded by the magnitude of the feature direction on the line. RNNs also appear to be implementing some form of spatial superposition in some cases, partitioning the activation space for several different features; this behavior clearly contrasts from Figure 5, which, for many input features and delays of up to 10, would only choose to represent one feature. However, we also often see a collapse of several feature directions onto the same line. In fact, although we study up to 7 input features, the models typically converge to approximately 2-3 principle directions. We can also see how for 2 features, most RNNs learn to represent these features approximately orthogonally. "
> >
> > Remarkably, in the SSM with high sparsity, we exactly recover the pentagon of 5 features characteristic of spatial superposition. We suspect that the SSMs geometric strategies are a result of each feature direction placing itself in the interference-free space of the other feature directions' readouts (hence the spiraling, perpendicular, and pentagon shapes).”
> >
> > As we stated in our response to Reviewer Jqdv, we believe that the generalizable insight from our work is that superposition (non-orthogonality of feature representations) occurs in capacity-constrained RNNs and is uniquely affected by both “space” and timing/memory demands, which interact and present representational tradeoffs, some of which we characterize here. Furthermore, we show that temporal superposition is critically a result of representing time– indicating that time-dependent input and output drive this behavior. It is our hope that these ideas can be built on to better predict and understand the mechanisms and representational strategies employed by RNNs in diverse task settings.
> >
> > > I find that the authors are less direct with how far RNN studies have come (specifically, the low-rank RNN studies). In many cases, we can now study the low-dimensional subspaces learned by RNNs and draw the exact flow maps they learn (not just the feature dimensions). In that sense, more modern discussion of low-rank RNNs is desirable.
> >
> > We thank the reviewer for addressing this issue and apologize for our omission. It was not our intention to underrepresent this work and we are very excited by the recent progress in low-rank RNNs. We have now added the following text in the related work section:
> >
> > “One important line of theoretical work has studied low-rank RNNs (Mastrogiuseppe & Ostojic, 2018; Schuessler et al., 2020a; Beiran et al., 2020; Dubreiul et al., 2022). These interpretable models have low-dimensional recurrent dynamics, allowing their exact phase portraits to be visualized; furthermore, these dynamics can be directly related to the underlying connectivity statistics. Related to our work, low-rank connectivity also acts as a form of capacity constraint, although the effects of such constraints have not been studied explicitly (but see Beiren et al. (2023) for comparison between low-versus-full-rank RNNs).”

---

> > > ### Author Response · Authors · 2025-11-27
> > > **Author Response Continued (2)**
> > >
> > > > It is unclear how many RNNs have developed these representations (especially for nonlinear ones) and how many failed to learn or learned completely different solutions. More rigor in reporting is desirable.
> > >
> > > We thank the reviewer for pointing this out and apologize for not having reported this in the paper. When reporting the RNNs that have developed these representations, we are always showing models that achieved the best performance to demonstrate the optimal learned solution. We note that optimizing with a two-dimensional hidden space is challenging for gradient descent and that not all models reach this solution. In fact, in the original paper on toy models of superposition (Elhage et al., 2022), the authors also study a 2-dimensional hidden state and report that they fit each model multiple times and take the solution with the lowest loss due to these optimization challenges. Indeed, superposition should be more efficient in higher dimensional spaces because there are many more “almost orthogonal” directions (Johnson-Lindenstrauss lemma) and with sparse features, compressed sensing shows that you can pack far more features than dimensions with tolerable interference.
> > >
> > > We have now run a new experiment where we train 1000 models of each architecture (linear, SSM, nonlinear) for various delay ($k=2,3,5,7$) and sparsity ($1-p=0.9, 0.97, 0.99$) values, and use architecture-specific metrics to quantify whether models exhibit characteristics of the predicted optimal geometry. We report the percentage of models that achieve values indicative of our predicted optimal geometry in Table 1 in Appendix I.1. In linear RNNs, the number of models that achieve the expected spiral sink decreases with increasing $k$ (from ~25-30% for $k=2$ to ~11-15% for $k=7$), as the task simply becomes too challenging for the simple architecture to learn well. In many cases, linear RNNs resort to oscillatory behavior which achieves suboptimal loss. In contrast, over 40% of SSMs trained on larger $k$ and under high sparsity learn the expected spiral sink + interference-free space solution. Finally, in nonlinear RNNs, about ~25-30% learn to fully exploit the interference-free space for $k=2$, decreasing for larger $k$.
> > >
> > > **Questions**
> > > > Eq. (1) is actually the more common RNN architecture in computational neuroscience. Hence, Appendix F.1. is making incorrect claims. Also, the connection of Eq. (1) to other form of vanilla RNN is well known. See [1].
> > >
> > > We thank the reviewer for highlighting this and apologize for the confusion. In Appendix F.1, by “traditional RNN architecture,” we were referring to classic RNN architectures used in machine learning (for example, Jordan and Edelman networks), which apply the activation to the summation of the input and recurrent inputs. You’re absolutely right that the architecture we use is typical in computational neuroscience. We have now modified Appendix F.1 to detail that this is different from the classic ML RNN, but identical to RNNs used in computational neuroscience. Thank you also for pointing us to this reference which we were unaware of. We have now included it in Appendix F.1.
> > >
> > > > Please emphasize in the abstract that your main contributions hold exactly for linear RNNs, and then you show that some of the insights do generalize to nonlinear counterparts. However, generalization beyond the particular task is not shown, please state this explicitly as well.
> > >
> > > We apologize for not making this more clear in the abstract. We note that we also provide an analytical approximation of the expectation of the loss for RNNs with ReLU readout in the limit of high sparsity. We have now modified the abstract to the following (new text in **bold**):
> > >
> > > “We develop a theoretical framework **in RNNs with linear recurrence trained on a delayed serial recall task** to better understand how properties of the data, task demands, and network dimensionality lead to different representational strategies, **and show that these insights generalize to nonlinear RNNs**.”
> > >
> > > > What makes feature A different than others in Figure 4?
> > >
> > > In Figure 4, we weight the loss by a different magnitude scalar (“importance”) for each feature (A being the largest at 1, E being the smallest at $0.97^4$), as is done in Elhage et al., 2022. This is mostly done for visualization purposes (so that we can control the ordering of the features ‘dropping out’). We emphasize that the pattern in Figure 4 still occurs if features are weighted equally– just that the remaining feature (like feature A) is random out of the 5 possible features.

---

> > > > ### Author Response · Authors · 2025-11-27
> > > > **Author Response Continued (3)**
> > > >
> > > > > Figure 5 is very interesting, why is it in Appendix? I would argue it may be the most interesting figure in this paper.
> > > >
> > > > Thank you for your interest! We didn’t include this figure in the main text mostly due to space constraints, favoring other figures that focused solely on feature geometry. We find the learning dynamics of feature geometry very interesting and relevant, and we’ve now moved this figure into the main text.
> > > >
> > > > > Are you aware of [2]? If so, how does your work, especially Fig. 5, compare to their findings? In my reading, it seems you also find that geometric restructuring (as Haputhanthri et al. 2024 has defined it) can happen even without emergence of attractors, as your case simply uses a spiral. That seems like a noteworthy connection/extension of prior work as well.
> > > >
> > > > We thank the reviewer for pointing us to this reference. We were not previously aware of this paper and have now included it in our discussion of learning dynamics in relation to Figure 5. Interestingly, the findings in Haputhanthri et al., 2024 seem very related to the concept of saddle-to-saddle dynamics that has been studied in feedforward networks (e.g., Jacot et al., 2022). From our perspective, it seems reasonable that “geometric restructuring” may not necessarily depend on attractors. In particular, the abrupt staircase-like drops in the loss curve have been shown to correspond to learning of the data correlation eigenvalues (e.g., Saxe et al., 2014; Proca et al., 2025), which may correspond to attractors or other configurations, like the spiral we see here. Indeed, we find it very interesting to link the learning dynamics of the connectivity parameters/eigenvalues to RNN’s underlying geometry.
> > > >
> > > > References:
> > > >
> > > > Jacot et al. Saddle-to-saddle dynamics in deep linear networks: Small initialization training, symmetry, and sparsity. *Arxiv*, 2022.
> > > >
> > > > Saxe et al. Exact solutions to the nonlinear dynamics of learning in deep linear neural networks. *ICLR*, 2014.
> > > >
> > > > Proca et al. Learning dynamics in linear recurrent neural networks. *ICML*, 2025.
> > > >
> > > > > My overall assessment is as follows: This work is limited in ambition and has some issues with rigor, both of which limit its contributions. However, it does introduce an interesting toy model and compared to how far we have come with RNN analyses, it does show how rudimentary the superposition/interpretability ideas in LLMs are. We need to do a lot better. With correct placement into the RNN literature, in which we can now study high-dimensional RNNs and their learned algorithms exactly (see recent low-rank RNN studies), I think this manuscript can be a fun read for the ICLR audience.
> > > >
> > > > We share the reviewer’s critique about the need to improve our understanding of superposition/interpretability. Indeed, there are many open problems in the field (Sharkey et al., 2025), and our aim in this work was to expand the understanding of this behavior in a new architecture with different constraints that affect geometry. Despite the existing limitations in the field of mechanistic interpretability, we are encouraged by the growing work that indicates it is a useful framework under which to understand and intervene on highly complex models (Templeton et al., 2024; Gao et al. 2024; Lindsey et al., 2025). We agree that the theoretical work in RNNs has been truly impressive over the past few years, and we hope that by applying the ideas from mechanistic interpretability in RNNs, we might further our understanding of how task demands, capacity, and sparsity shape geometry and behavior, and provide a useful (toy) framework under which to study RNNs.
> > > >
> > > > References:
> > > >
> > > > Sharkey et al. Open Problems in Mechanistic Interpretability. *ArXiv*, 2025.
> > > >
> > > > Templeton, et al. Scaling Monosemanticity: Extracting Interpretable Features from Claude 3 Sonnet. *Transformer Circuits Thread*, 2024.
> > > >
> > > > Gao et al. Scaling and evaluating sparse autoencoders. *ICLR*, 2025
> > > >
> > > > Lindsey et al. On the Biology of a Large Language Model. *Transformers Circuits Thread*, 2025.

---

### Official Review · Reviewer_Jqdv · 2025-11-01

**Soundness:** 3
**Presentation:** 4
**Contribution:** 3
**Rating:** 8
**Confidence:** 4

**Summary:**

The manuscript addresses the question of how artificial recurrent networks can store information, specifically in settings when the dimensionality of the state space is smaller than the total number of features that needs to be represented. Specifically, the authors investigate a phenomenon similar to that of spatial superposition, but in the temporal domain. They find that, when inputs are temporally sparse, even small recurrent neural networks can represent features over a comparatively long time by packing them into a subspace of the dynamics that is orthogonal to the network output, and by forgetting inputs once they become irrelevant. The manuscript focuses on a rather simple task in very simple RNN (linear and non-linear, low-d) to achieve a thorough understanding of the underlying mechanisms, combining analytical insights and simulations.

**Strengths:**

The paper is well motivated and well written. Relevant literature and the implications of the work are discussed appropriately. The paper is technically strong.

The focus on simple, small RNN, and on a single simple task, allows the authors to achieve a deep understanding of the underlying mechanisms. In particular, the authors combine simulations with insights from analytical derivations, which together make a very convincing case for their findings and conclusions.

The paper makes an interesting conceptual advance towards understanding the basic principles by which recurrent neural networks operate, by identifying a phenomenon akin to spatial summation in the temporal domain.

**Weaknesses:**

The focus of the paper on very simple, small RNNs and a single, comparatively simple task allows the authors to achieve a deep understanding of the underlying mechanisms, but also raises some questions about the broader relevance of the resulting insights. The main weakness may be the exclusive focus on the k-delay task. Many, if not all, findings about temporal superposition presented in the paper seem to be a direct consequence of using this task, which requires RNNs to remember an input feature for a fixed number of time-steps, then output it, and finally forget it. The authors make a strong case that that the type of dynamics they observe, which involves their mechanisms of temporal superposition, is “optimal” for this task. In fact, the mechanism they describe “makes a lot of sense” given what we know about RNN trained on simple tasks and is arguably not entirely surprising. It is less clear what are the implications of their findings to other tasks, which may require RNN to maintain information for a variable amount of time (variable k, whereby e.g. the RNN output is interrogated by providing a dedicated “go” signal) or to do more than just remember an input feature (e.g. by producing outputs that could control simple movements).

In particular, I would expect that for a variable k (i.e. a randomized time between input and output) the type of solutions implemented by the RNNs are very different. In simple neuroscience decision-making tasks, RNN trained with fixed delays are known to produce rotational dynamics that is timed just right to put their activity into the right location of state space when say an output is needed, which is exactly what the authors find. On the same tasks, training with random delays instead leads to stable representations during the delay period (based e.g. on stable fixed points). In such a setting, temporal summation does not seem to be relevant, and instead of spatial summation is at play.

**Questions:**

The authors should discuss or explore the implication of their findings for other tasks, specifically also tasks that involve variable, randomized delays, whereby RNN outputs are triggered by the onset or offset of a dedicated go cue.

The authors should relate their work to past studies with RNN that have found dynamics that seem closely related to what is shown in this manuscript.

I found the description of the phase transition hard to follow, only few results are shown to validate the point. The corresponding section is less clear and weaker than the remainder of the paper.

---

> ### Author Response · Authors · 2025-11-27
> **Author Response to Reviewer Jqdv**
>
> Thank you for your review and your thoughtful comments. We are delighted that you found our paper strong and an interesting advance.
>
> **Weaknesses**
>
> **Exclusive focus on k-delay task and implications for other task settings.**
> We thank the reviewer for raising this important point. Indeed, in this work we focus on the $k$-delay task, which, although perhaps simple, has a few advantages. (1) The $k$-delay task provides a direct way to control memory demand (through the hyperparameter $k$), accounting for time-dependent inputs and outputs. Indeed, It has been used in seminal papers to study memory capacity in RNNs (Jaeger, 2002). (2) The $k$-delay task is easier to work with analytically. Furthermore, the $k$-delay task can be performed by both linear and nonlinear networks (unlike cued random delay which requires nonlinearity), allowing us to study and compare between solutions found in different architectures. (3) The $k$-delay task is also the direct extension of the task considered in Elhage et al., 2022 to the temporal domain. Hence, as we show in Section 4.5, we can even interpolate between spatial and temporal superposition by changing $k$. Being the first paper to introduce the concept of temporal superposition, we wanted to focus on a task that would allow us to compare and contrast with spatial superposition introduced in Elhage et al., 2022.
>
> We fully agree with the reviewer that it is important to consider more complex task settings and how superposition may generalize to them. Indeed, it is almost certain that the geometric strategies between different tasks, with randomized delay or manipulation of input will differ. We do not claim that the feature geometry studied here will apply to every other task setting in RNNs. Instead, we believe that the generalizable insight from our work is that superposition (non-orthogonality of feature representations) occurs in capacity-constrained RNNs and is uniquely affected by both “space” and timing/memory demands, which interact and present representational tradeoffs, some of which we characterize here. Furthermore, we show that temporal superposition is critically a result of representing time– indicating that time-dependent input and output drive this behavior. It is our hope that these ideas can be built on to better understand the mechanisms and representational strategies employed by RNNs in diverse task settings.
>
> References:
> Jaeger. Short term memory in echo state networks. German National Research Institute for Computer Science, 2002.
> Elhage et al. Toy models of superposition. Transformers Circuit Thread, 2022.
>
> **Stable fixed point solution to tasks with random delays.**
> We thank the reviewer for raising this important and nuanced point! Indeed, before our experimentation (see below), we expected a similar phenomenon to happen if our model is tasked with a random-delay– that feature directions would no longer encode “age”/time through rotations, but instead stabilize in one direction (as a fixed point) that can be read-out at some variable time, such that $w_{s} \approx w_{s’}$ for $s \neq s’$. This would recover something similar to pure spatial superposition where, for example features A-E are represented simultaneously in separate directions but there is no time-related component. We have now run new experiments with random-delay, and we see something somewhat similar to this prediction. While feature directions tend to stabilize along a line, age is still partially encoded by the scale of the feature direction, which shrinks towards the origin over time. We suspect that the models we study do not develop persistent/stable fixed point solutions in this setting due to the sequential ordering of inputs which is still time-dependent, and perhaps because of their limited expressivity.
>
> Your point precisely highlights how temporal superposition is a result of time-dependence: RNNs represent time (both input timing and output timing) through feature directions. When timing is less important (e.g., with random delay or without ordered stimuli/sequence), these representations of time may be less pronounced or disappear, recovering spatial superposition. In this sense, removing time-dependence from the task encourages the RNN to act similarly to a feedforward network (through fixed point solutions).

---

> > ### Author Response · Authors · 2025-11-27
> > **Author Response Continued**
> >
> > **Questions**
> > > The authors should discuss or explore the implication of their findings for other tasks, specifically also tasks that involve variable, randomized delays, whereby RNN outputs are triggered by the onset or offset of a dedicated go cue.
> >
> > Thank you for your helpful suggestion. In order to address this issue, we have now included new experiments across all architectures where RNNs are tasked with performing a sequence reproduction with cued random delay in Appendix H with figures. We copy the text here for convenience:
> >
> > “In this paper we primarily focus on tasks with a fixed $k$-delay. Here, we instead consider the effect of training on a task with random delay. The task we consider is identical to the $k$-delay task in that the RNN must reproduce the input sequence after $k$ timesteps, but now $k$ is random for each training sample ($k\sim\text{Uniform}(0,10)$. One dimension of the input corresponds to the cue, which remains 0 until the randomly selected $k$, after which it is set to 1 and the RNN is tasked with outputting the sequence, corresponding to the input from the $t-k, \forall t>k$. We train RNNs of each architecture (linear: Figure 13, SSM: Figure 14, nonlinear: Figure 15) and visualize how the feature geometry changes as the number of input features is varied (the rows) and sparsity is varied (the columns).
> >
> > Although we are cautious about overinterpreting these plots, we provide a preliminary analysis. The results seem to suggest some intermediate geometry between spatial superposition and temporal superposition. Indeed, the notion of time-dependency here marks a departure from the rest of the paper in that the \textit{sequential ordering} of features is important for the task, but a \textit{time-dependent} output is not. We see that most RNNs form solutions where feature directions lie on a shrinking line (instead of a spiral sink), with a fixed point at the origin (corresponding to 'forgetting'). 'Age' (for sequential ordering) is still partially encoded by the magnitude of the feature direction on the line. RNNs also appear to be implementing some form of spatial superposition in some cases, partitioning the activation space for several different features; this behavior clearly contrasts from Figure 5, which, for many input features and delays of up to 10, would only choose to represent one feature. However, we also often see a collapse of several feature directions onto the same line. In fact, although we study up to 7 input features, the models typically converge to approximately 2-3 principle directions. We can also see how for 2 features, most RNNs learn to represent these features approximately orthogonally.
> >
> > Remarkably, in the SSM with high sparsity, we exactly recover the pentagon of 5 features characteristic of spatial superposition. We suspect that the SSMs geometric strategies are a result of each feature direction placing itself in the interference-free space of the other feature directions' readouts (hence the spiraling, perpendicular, and pentagon shapes).”
> >
> > > The authors should relate their work to past studies with RNN that have found dynamics that seem closely related to what is shown in this manuscript.
> >
> > We thank the reviewer for pointing out our omission of relevant literature. We have now included the following text in the related work section on RNNs:
> >
> > “Similar to the feature geometry we see here, previous work in RNNs has observed rotational dynamics/sequential activity (Rajan et al., 2016; Orhan & Ma, 2019; Cueva et al, 2020; Zhang et al., 2021) for tasks with fixed delay, thought to encode temporal information. Moreover, other work has shown that RNNs trained on tasks with random delays instead exhibit persistent activity, in the form of fixed point attractors (Orhan & Ma, 2019; Liu et al., 2021; Xie et al., 2022b), similar to pure spatial superposition (for example, as we see when $k=0$). Related to the interference-free space we study in our model, RNNs trained in motor-preparation paradigms similarly develop output-null subspaces where intermediate preparatory activity does not affect behavior (Schimel et al., 2024).”
> >
> > > I found the description of the phase transition hard to follow, only few results are shown to validate the point. The corresponding section is less clear and weaker than the remainder of the paper.
> >
> > We thank the reviewer for the valuable feedback and apologize for the lack of clarity on our part. We have now included additional plots in the phase transition figure to provide visualizations of the changing geometry as sparsity is varied. Additionally, we have edited the text in an effort to make the writing more clear. In Figure 9, we also include a grid of feature geometry as sparsity and memory demand ($k$) is varied which illustrates this more.

---

### Author Response · Authors · 2025-11-27
**Global Response**

We would like to thank the area chair and the reviewers for their thoughtful work in reviewing our paper. We appreciate the positive feedback on the relevance and potential of the work for the mechanistic interpretability and computational neuroscience community, as well as the encouraging comments about the presentation quality and technical soundness of our paper. We are very grateful for the constructive comments and feedback that will enable substantial improvement to the manuscript.

While the reviewers found the results “very interesting... of broad relevance” (reviewer pihS), “fun to read” (DPAE), “significant” (iPFA), and “technically strong” (Jqdv), they also identified improvements that can be made to broaden the impact and relevance of the toy model setting we consider. The reviewers unanimously agreed on the need to generalize our analysis (in which we study fixed-delay serial recall) to tasks with random delays. Additionally, reviewers DPAE and pihS expressed concerns about the small sizes of the RNNs we studied. We also appreciate the reviewers’ suggestions on restructuring some of the results and for pointing us to relevant literature on RNNs and neural dynamics to include. We discuss the insightful comments received in the responses to individual reviewers. To address the reviewers’ feedback, we have taken the following actions:
1. We have run new experiments on tasks with random delays across all architectures (Appendix H).
2. We have run new experiments with larger RNNs (of hidden size 100) and created metrics to quantify whether the largest feature directions group into an interference-free space and whether the output feature direction aligns with the readout at the appropriate time (Section 4.5 and Appendix G).
3. We have generalized our derivation of the expectation of the loss to vector inputs and outputs (Appendix G.1).
4. We have added discussions of relevant literature suggested by the reviewers to the manuscript.
5. We have incorporated more of the results in our Appendix into the main text. In particular, we have now included the figure on learning dynamics, discussed metrics to quantify superposition in higher dimensional RNNs, and provided visualizations of changing feature geometry occurring in the phase transition figure.

Changes to the manuscript are indicated by the red colored text.

Again, we thank the reviewers for their time and insightful comments. We hope our responses below will address any other concerns and questions raised.

---

### Meta-Review · Area_Chair_xB4j · 2026-01-05

**Summary:**

The paper introduces and analyzes “temporal superposition” in recurrent neural networks: how capacity constraints interact with time-dependent inputs/outputs to yield non-orthogonal feature representations across time. The work combines an analytically tractable linear RNN setting with extensive geometric visualizations and follow-up experiments in nonlinear RNNs/SSMs. The aim is to connect mechanistic interpretability ideas (superposition/interference) to recurrent dynamics.

Overall, the reviews are positive.
Three reviewers recommend clear acceptance (Jqdv, iPfa, pihS: 8/10). They highlight strong exposition, compelling conceptual framing, and a technically sound decomposition of interference terms that explains learned geometry.
Reviewer DPAE is more cautious (6/10). Main critique is limited ambition (single task; very small RNNs), missing placement within modern RNN/low-rank literature, and insufficient rigor regarding variability / prevalence of learned solutions. However, many of these concerns have been addressed in the rebuttal.

**Reviewer Concerns:**

The rebuttal addresses the main concerns:

- Generalization beyond fixed delay: authors added experiments on random/cued delay across architectures (Appendix H), showing an intermediate geometry between temporal and spatial superposition and discussing when fixed-point–like strategies emerge. This directly responds to the central “fixed k-delay only” (raised by Jqdv, also implicitly by DPAE (single task)).

- Scaling beyond 2D hidden states: authors added higher-dimensional analyses and new experiments with larger RNNs (e.g., hidden size 100, many features), along with metrics that quantify “interference-free space” usage and time-aligned readout (Section 4.5 / Appendix G). While still not “realistic scale,” this reduces the concern that the phenomena are artifacts of $N_h=2$.

 - Rigor / reporting variability: authors now clarify the selection procedures (best-of-N training) and added a large-scale training sweep (1000 seeds per architecture) reporting the fraction of runs that match predicted geometry (Appendix I.1). They also revise some figures to show mean/error over more runs (best 50 rather than best 5). (DPAE)

- Positioning within RNN literature: authors expanded the related work to include low-rank RNNs and prior dynamical observations (rotational dynamics vs persistent activity under random delays), and corrected a noted confusion about “traditional” RNN formulations in Appendix F.1.

After rebuttal, the main remaining limitations are (i) the analysis is still anchored to a toy delayed recall setting with limited exploration of manipulation/context-dependent computations, and (ii) the strongest theory is exact for linear recurrence, with nonlinear results primarily empirical/phenomenological. However, these limitations are now stated more explicitly, and the added experiments support the claim that the core geometric mechanisms extend beyond the narrowest initial setting.

Given the technical quality, clarity, novelty of framing temporal superposition, and the strong rebuttal that addresses most of the weaknesses, I recommend accepting the paper. The paper should be valuable to both mechanistic interpretability and computational neuroscience audiences as a careful, interpretable bridge between superposition-style thinking and recurrent dynamics. While Orals are very competitive at ICLR, this paper hits the bar.

**Reviewer Scores:**

- DPAE: Rating: 6;
either stayed or raised. The rebuttal was taken very seriously, but I think the reviewer would have stayed at 6 (suggesting weak acceptance) without going to 8.
All other viewers had rating 8. All of their concerns/comments were addressed, but still I think the reviewers would have kept their score.

---

### Decision · Program_Chairs · 2026-01-26

Accept (Oral)